# Protection of the *C. elegans* germ cell genome depends on diverse DNA repair pathways during normal proliferation

**Bettina Meier**[1☯], **Nadezda V. Volkova**[2☯], **Ye Hong**[1¤a], **Simone Bertolini**[1], **Víctor González-Huici**[1¤b], **Tsvetana Petrova**[1¤c], **Simon Boulton**[3], **Peter J. Campbell**[4,5,6], **Moritz Gerstung**[2,7]*, **Anton Gartner**[1,8,9]*

**1** Centre for Gene Regulation and Expression, University of Dundee, Dundee, Scotland, **2** European Molecular Biology Laboratory, European Bioinformatics Institute, Hinxton, United Kingdom, **3** Francis Crick Institute, London, United Kingdom, **4** Cancer, Ageing and Somatic Mutation Program, Wellcome Sanger Institute, Hinxton, United Kingdom, **5** Department of Haematology, University of Cambridge, Cambridge, United Kingdom, **6** Department of Haematology, Addenbrooke's Hospital, Cambridge, United Kingdom, **7** European Molecular Biology Laboratory, Genome Biology Unit, Heidelberg, Germany, **8** Department of Biological Sciences, School of Life Sciences, Ulsan National Institute of Science and Technology, Ulsan, Republic of Korea, **9** Center for Genomic Integrity, Institute for Basic Science, Ulsan, Republic of Korea

☯ These authors contributed equally to this work.
¤a Current address: Shandong Provincial Key Laboratory of Animal Cell and Developmental Biology, School of Life Sciences, Shandong University, Qingdao, China
¤b Current address: Institute for Research in Biomedicine, Barcelona, Spain
¤c Current address: MRC Protein Phosphorylation and Ubiquitylation Unit, University of Dundee, Dundee, Scotland
* tgartner@ibs.re.kr (AG); moritz.gerstung@ebi.ac.uk (MG)

**Data Availability Statement:** Sequencing data and variant calling files are available under ENA Study Accession Numbers ERP000975 and ERP004086. ENA sample IDs are annotated in S1 Table. R code

## Abstract

Maintaining genome integrity is particularly important in germ cells to ensure faithful transmission of genetic information across generations. Here we systematically describe germ cell mutagenesis in wild-type and 61 DNA repair mutants cultivated over multiple generations. ~44% of the DNA repair mutants analysed showed a >2-fold increased mutagenesis with a broad spectrum of mutational outcomes. Nucleotide excision repair deficiency led to higher base substitution rates, whereas *polh-1*(Polη) and *rev-3*(Polζ) translesion synthesis polymerase mutants resulted in 50–400 bp deletions. Signatures associated with defective homologous recombination fall into two classes: 1) *brc-1*/BRCA1 and *rad-51*/RAD51 paralog mutants showed increased mutations across all mutation classes, 2) *mus-81*/MUS81 and *slx-1*/SLX1 nuclease, and *him-6*/BLM, *helq-1*/HELQ or *rtel-1*/RTEL1 helicase mutants primarily accumulated structural variants. Repetitive and G-quadruplex sequence-containing loci were more frequently mutated in specific DNA repair backgrounds. Tandem duplications embedded in inverted repeats were observed in *helq-1* helicase mutants, and a unique pattern of 'translocations' involving homeologous sequences occurred in *rip-1* recombination mutants. *atm-1*/ATM checkpoint mutants harboured structural variants specifically enriched in subtelomeric regions. Interestingly, locally clustered mutagenesis was only observed for combined *brc-1* and *cep-1*/p53 deficiency. Our study provides a global view of how different DNA repair pathways contribute to prevent germ cell mutagenesis.

for the analysis of mutation rates and signatures as well as mutational clustering is available on GitHub under http://github.com/gerstung-lab/mutationaccumulation.

**Funding:** This work was supported by: AG: Wellcome Trust [COMSIG consortium grant RG70175 and Senior Research Award 090944/Z/09/Z] https://wellcome.org/. AG: Worldwide Cancer Research [18-0644] https://www.worldwidecancerresearch.org/. AG: Korean Institute for Basic Science [IBS-R022-A2-2020] https://www.ibs.re.kr/eng.do. The funders had no role in study design, data collection and analysis, decision to publish, or preparation of the manuscript.

**Competing interests:** The authors have declared that no competing interests exist.

## Introduction

Germ cells are required to pass genetic information from one generation to the next, rendering the maintenance of their genetic integrity particularly important. While germ cell mutations are the basis of evolution, mutational events tend to be detrimental and are associated with both reduced fitness and inherited disease.

Endogenous mutagenesis can be caused by nucleotide mis-incorporation during replication and by reactive cellular metabolites. Hydrolytic reactions trigger abundant depurinations, depyrimidinations, and the deamination of cytosine and 5-methylcytosine (for review [1]). Reactive oxygen species, byproducts of oxidative phosphorylation and oxygen-dependent enzymatic processes, induce 10,000–100,000 DNA lesions per cell per day, including base modifications such as 8-oxo-dG, thymine glycol and DNA single-strand breaks [2]. In addition, enzymatic and non-enzymatic mechanisms lead to base methylations. For instance, 3-methyl-adenine and 3-methyl-cytosine can lead to mutation by blocking replication, and $O^6$-methyl-guanine leads to G>A changes (for review [1]). Metabolic byproducts such as reactive aldehydes form DNA adducts that can crosslink bases from complementary DNA strands generating obstacles to replication and transcription.

DNA double-strand breaks (DSBs) are among the most toxic DNA lesions and arise when the replication fork is stalled by base modifications, repetitive DNA, DNA sequences prone to form tertiary structures, or collision with the transcription machinery [3]. Nevertheless, some cellular events require DSBs to be induced naturally, for example during immunoglobulin gene rearrangement to ensure immunoglobulin diversification. Additionally, during germ cell meiosis multiple DSBs are introduced in each chromosome, resulting in at least one crossover recombination event between homologs, thereby facilitating the exchange of genetic information and orderly chromosome segregation (for review [4]). Recombination requires a free DNA end to search for and invade a homologous DNA strand, which acts as a template to facilitate the restoration of genetic information. When DSBs occur in repetitive DNA such as tandem repeats or interspersed repeat elements like Line and Alu sequences, 'homology search' provides a formidable challenge.

Nevertheless, only a vanishingly small fraction of primary lesions leads to mutations, highlighting how effective DNA damage response mechanisms are in detecting and mending multifarious forms of DNA damage. The analysis of the observed mutations has the potential to shed light on the primary mutagenic lesion. When the amount of DNA damage introduced by a mutagenic process exceeds the capacity of DNA repair, distinct patterns of mutations, referred to as mutational signatures or spectra, arise. Here, we characterize genome-wide mutational spectra by analysing the number and distribution of single and multi-nucleotide variants (SNVs and MNVs), small insertions and deletions (indels) and structural variants (SVs) composed of larger (over 400 bp) deletions, inversions, duplications, and chromosomal translocations.

The rates and types of germline mutations were previously studied in humans and model organisms such as mouse, fruit fly, *C. elegans*, and primates [5–8]. These studies discovered a relatively uniform mutation pattern, sometimes referred to as 'signature 5' [6, 9], which has also been observed in different human somatic tissues. The underlying aetiology of this signature–and whether it is the product of a single, or multiple mutational processes–remains unclear. We recently reported a high-level analysis comprising 2700 *C. elegans* genomes treated with 11 genotoxic agents. ~40% of analysed DNA repair mutants exhibited altered mutations, thus highlighting a prominent role of DNA repair pathways in shaping mutation rates and signatures [10].

Here, we focus our analysis on mutations accumulating when *C. elegans* is propagated over generations in the absence of exposure to exogenous mutagens to enhance our understanding

of how mutagenesis is prevented in the germ cell lineage, largely using the same primary set of data. This study encompasses 528 genomes derived from wild-type and 54 single and 7 double DNA repair mutants. DNA repair mutants were chosen to encompass the majority of known DNA repair and damage response pathways. We provide a comparative, detailed and systematic genome-wide analysis of the contribution of these DNA repair pathways towards maintaining germ cell genome integrity.

## Results

### Mutation rates in *C. elegans* wild-type and DNA repair mutants

*C. elegans* offers a suitable system to study mutation accumulation (MA) over generations based on its short life-cycle of three to four days and its ability to self-fertilize. Self-fertilization enables the clonal propagation of lines from single animals randomly picked in each generation. In our mutation accumulation experiments (MA), we propagated (typically for 20 or 40 generations) 5–10 clonal lines for each genotype and randomly selected a minimum of 3 lines for sequencing (Fig 1A) (S1 Table for list and description of DNA repair mutant lines analysed). Out of 528 whole genome sequencing (WGS) datasets we analyse as part of this study (S1 Table), 472 were previously deposited (S1 Table) [10], and 56 (corresponding to 5 newly generated double and 6 single mutant strains) were newly deposited (S1 Table). DNA repair mutants were chosen to encompass the majority of known DNA repair and damage response pathways. Genomic DNA for sequencing was isolated from starved nematode populations, each a clonal expansion from a single L4 stage hermaphrodite from the first or last propagated generation (Materials and Methods, [10–12]). Crucially, these lines pass through a single-cell bottleneck provided by the zygote, enabling us to analyse how mutations arise in the germline (Fig 1A).

Calculating mutation rates from more than 30 wild-type MA lines including 5 lines grown for 40 generations, comparing mutations from the first and last generation (Fig 1A), we refined our previous mutation rate estimations [10–12] to ~0.9 mutations in the diploid *C. elegans* genome per generation (Materials and Methods). This corresponds to ~2.9 x $10^{-10}$ (95% CI: 2.5 x $10^{-10}$–3.2 x $10^{-10}$) mutations per base pair and germ cell division. Mutations were equally distributed across the wild-type genome with no evidence of clustering (Fig 1B). The most frequent mutations were a) single base insertions, with prevalent T>A changes in the context of a 5'A and a 3' T, and b) deletions in homopolymeric sequences (Fig 1C), indicative of replication slippage as a source of mutations in wild-type. Together with our previous data on MMR deficient strains [11] and the estimates provided below, this suggests that replication polymerase slippage in homopolymeric sequences is the most frequent type of genetic error.

Across 61 *C. elegans* DNA repair deficient mutants ([10–12], S1 Table), the median mutation rate was close to that observed in wild-type: 0.82 heterozygous base substitutions per generation compared to 0.57 (standard deviation SD = 0.04) in wild-type, 0.25 indels (0.26 (SD = 0.03) in wild-type) and 0.03 SVs (0.02 (SD = 0.01) in wild-type). However, mutation rates varied by mutation type, making the comparison of overall mutation rates misleading. We therefore stratified mutations into 1) single and multi-nucleotide variants, 2) indels smaller than 400 base pairs and 3) SVs (Fig 2, S1 Fig). Interestingly, 69% (42 out of 61) of DNA repair deficient strains displayed mutation rates significantly different from wild-type in at least one mutation class (false discovery rate FDR = 5%), with an over 2-fold change in 44% (27 out of 61) of mutants.

Besides the previously reported high mutation rate in DNA mismatch repair (MMR) mutants (*pms-2* and *mlh-1*), with 25–30 times more base substitutions and a ~100-fold increase in indels per generation [11, 13, 14], a ~2 fold increase of single SNV was observed in

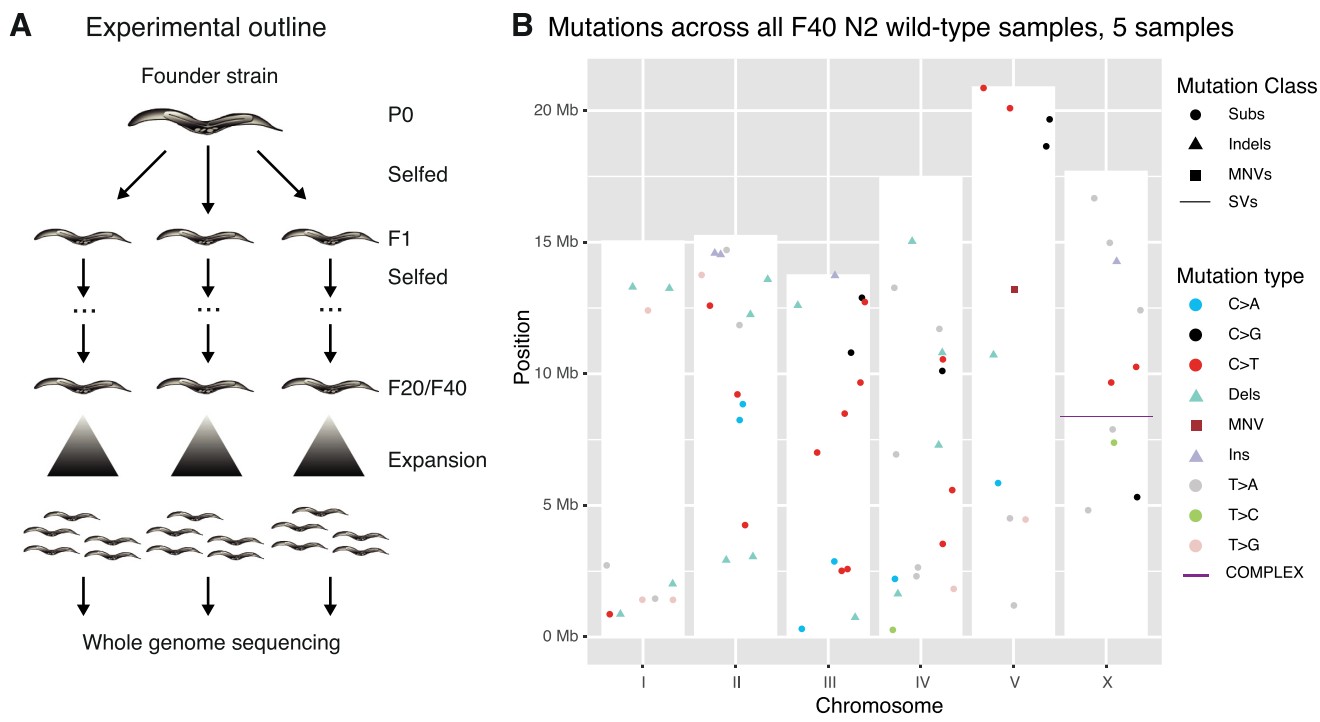

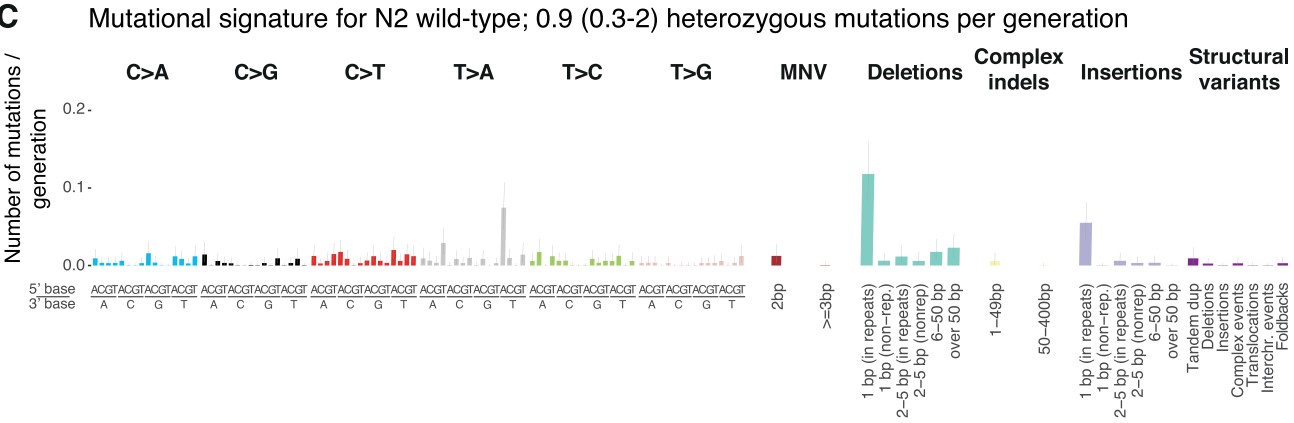

**Fig 1. Experimental outline and background mutagenesis in wild-type. A.** L4 larvae (F1 generation) from a parental founder strain (P0) were individually picked onto NGM plates and allowed to self-fertilize prior to picking individual L4 larvae of the next generation (F2) from each F1 plate. This process was repeated until clonal lines reached generation F20 or F40. Clonal lines were then allowed to expand, harvested, and prepared for whole genome sequencing (Materials and Methods). **B.** Mutation types and their location on the 6 *C. elegans* chromosomes (I-V and X) across all wild-type samples and mutation classes. The height of the white bars corresponds to the length of the respective *C. elegans* chromosome. Single nucleotide variants are indicated by a dot, dinucleotide variants (DNVs) by a square, indels divided in deletions (D) and insertions (I) by a triangle, and structural variants (SVs) by a line. **C.** Average number of heterozygous mutations in the N2 wild-type genome per generation across all mutation classes and types. Single nucleotide variants are shown in the context of their 5' and 3' base. Grey bars denote 95% credible intervals for the number of mutations in each type. "Complex indels" class denotes deletions with insertions. Data for N2 was previously shown in ([10] Fig 1C) . Information related to the 528 whole genome sequencing WGS primary-source datasets (56 deposited in this study, 472 deposited in (Suppl Data 1 and Supple Note 1 of [10] can be found in S1 Table).

several mutants defective for NER, HR, direct damage reversal (DR), and helicases (Fig 2, red dots). Moreover, ~3–5 fold more SNVs occurred in *smc-5* and *smc-6* HR mutants, as well as in the *cep-1; brc-1 brd-1* triple mutant defective for the *C. elegans* orthologs of BRCA1, its binding partner BARD1, and p53 (Fig 2, red dots). The DNA interstrand crosslink repair mutant dog-

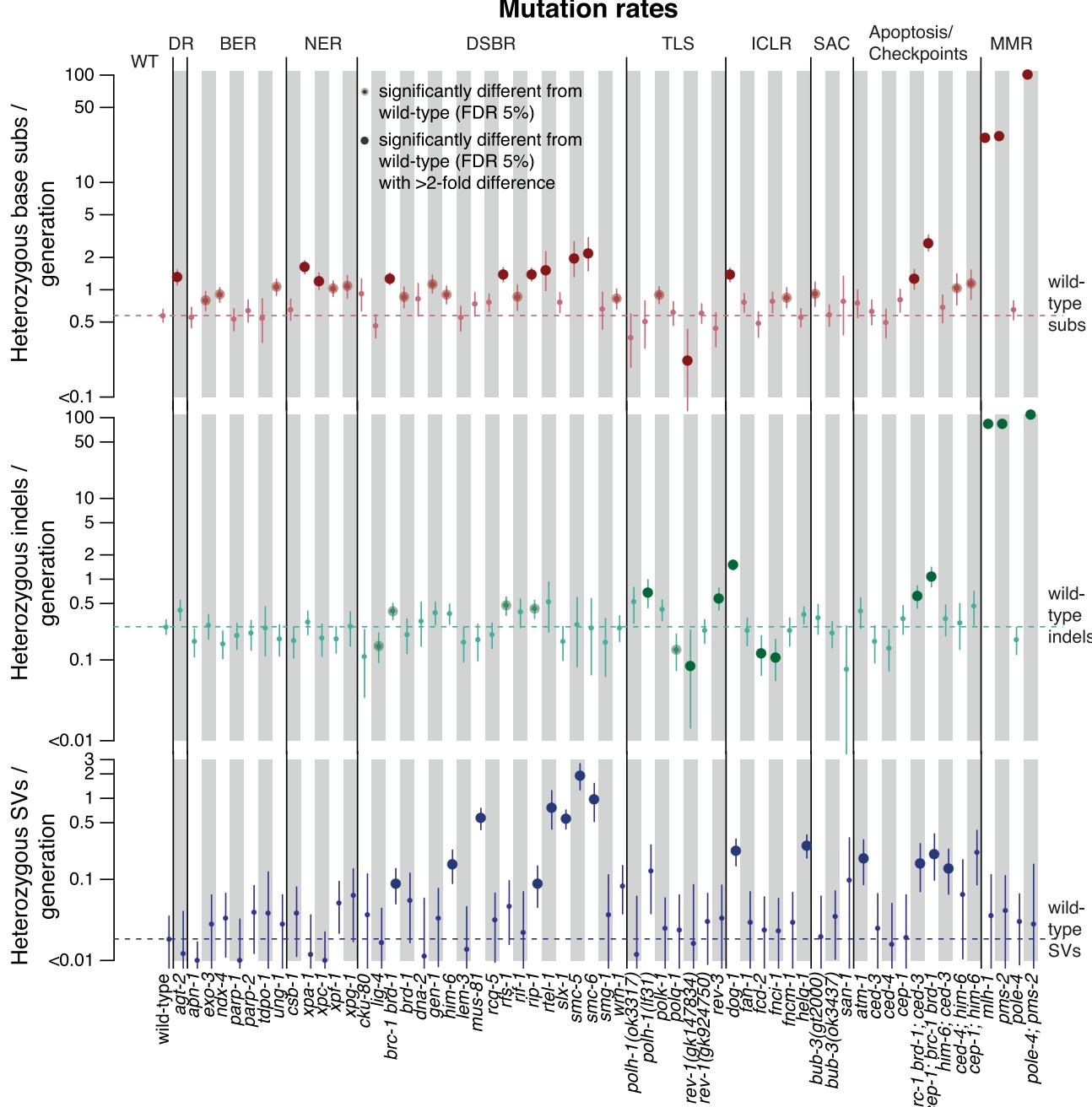

**Fig 2. Mutation rates across 60 *C. elegans* genotypes stratified by mutation type: Base substitutions, indels, and structural variants.** Mutation rates are shown as number of heterozygous mutations per generation for N2 wild-type (WT), and mutants used in this study grouped by the major DNA repair pathway they contribute to; direct damage reversal (DR), base excision repair (BER), nucleotide excision repair (NER), DNA double-strand break repair (DSBR), translesion synthesis (TLS), crosslink repair (ICLR), spindle assembly checkpoint (SAC), apoptosis, and mismatch repair (MMR). Base substitutions are shown in red (top), indels in green (center) and structural variants in blue (bottom). Dotted lines denote the mutation rates for wild-type. Error bars show the 95% confidence intervals; large dots represent variants with 2-fold increased or decreased mutation rates over N2 wild-type which are statistically significant with a false discovery rate (FDR) below 5%. All CIs which extend below the lower edge of the plot have zero as their lower border. Information related to the 528 whole genome sequencing WGS primary-source datasets (56 deposited in this study, 472 deposited in (Suppl Data 1 and Supple Note 1 of [10] can be found in S1 Table).

1/FANCJ accumulated ~6 times more indels compared to wild-type, and several HR and TLS deficient strains showed ~2 fold increased indel rates (Fig 2, green dots). Finally, the number of structural variants (SVs) tended to be elevated in lines compromised for HR and in various DNA helicase mutants (*dog-1*, *helq-1*, *him-6/BLM*, *rtel-1*) (Fig 2, blue dots).

Several DNA repair mutants, namely *fcd-2/FANCD2* and *fnci-1/FANCI* DNA ICL-repair defective lines, and the microhomology mediated end-joining (MMEJ) defective *polq-1/POLQ* mutant exhibited reduced indel rates (Fig 2, green dots) Additionally, *polq-1* mutants harboured reduced SNVs (Fig 2, red dot). POLQ-1 dependent MMEJ is an error prone pathway, in which resected 3' single-stranded overhangs pair at their complementary terminal nucleotide(s) to prime DNA synthesis, often leading to small deletions [15–18]. However, given that indel and SV mutation rates in our dataset are already low in wild-type and more wild-type than mutant samples were included in the analysis (wild-type n = 30, mutant n = 4–8) the sample variance in genotypes with mutation rates close or lower to wild-type may be underestimated. We therefore caution that the observed reductions in mutagenesis levels are likely to be false discoveries.

## Direct damage reversal (DR), base excision repair (BER), nucleotide excision repair (NER), and single-strand break (SSB) repair

We next wished to systematically characterise the signatures and features of mutations accumulated over generations by DNA repair pathways. Mutants deficient in DR, BER, and NER did not show large changes in the overall mutation spectra, but several small differences in particular mutation types (Fig 3A, S2A Fig).

In addition to AGT-1, which facilitates direct damage reversal by removing methyl moieties from $O^6$-methyl guanine, AGT-2 encodes for a further predicted *C. elegans* $O^6$-alkylguanine DNA alkyltransferase [19]. We found a 2-fold increased mutation rate in *agt-2* deficient lines (Fig 2), owing to an elevated frequency of C>T changes caused by the mispairing of $O^6$-methyl guanine with T (Fig 3A and 3B). Interestingly, *agt-2* mutants exhibited a moderate degree of mutation clustering, evidenced by 7 cases of 2–3 mutations located in closer proximity to each other than expected by chance, scattered across 10 *agt-2* mutant lines (Fig 3C, S2B Fig). We also confirmed increased numbers of C>T changes in mutants defective for *ung-1*, an Uracil-DNA glycosylase that excises uracil during BER [20] (Fig 3A and 3B). Uracil is introduced via UTP mis-incorporation or cytosine deamination and pairs with adenine, which leads to C>T mutations. Other BER mutants, including mutants deficient in PARP-1 and PARP-2, the two *C. elegans* poly-ADP ribose polymerases needed for SSB repair, did not show altered mutation rates compared to wild-type (Fig 2, S2A Fig).

The NER pathway is involved in the repair of bulky DNA adducts and DNA crosslinks, both of which cause a structural distortion of the DNA double helix [21]. *xpa-1*, *xpf-1*, and *xpg-1* lines compromised for all NER and *xpc-1* lines solely defective for global genome NER (but not *csb-1* lines uniquely defective for transcription coupled NER) showed increased mutation rates without overt changes in mutational signatures (Figs 2, 3A and 3B, S3A Fig). We speculate that this increased mutagenesis might be caused by cyclopurines induced by reactive oxygen species, and/or from exposure to ambient and fluorescent light. Comparison between the *C. elegans* NER signature adjusted for the human nucleotide composition (see [11]) and COSMIC signatures SBS8 (associated with NER using human organoids) and SBS5 (associated with NER defects in urothelial cancers) showed no conformity (cosine similarity scores 0.64 and 0.56, respectively) (S3B Fig). DSB repair by nonhomologous DNA end-joining (NHEJ), microhomology mediated end-joining (MMEJ) and homologous recombination (HR).

DSB repair is facilitated by several redundant pathways. HR is considered largely error-free, restoring genetic information using an intact homologous DNA strand as a repair template.

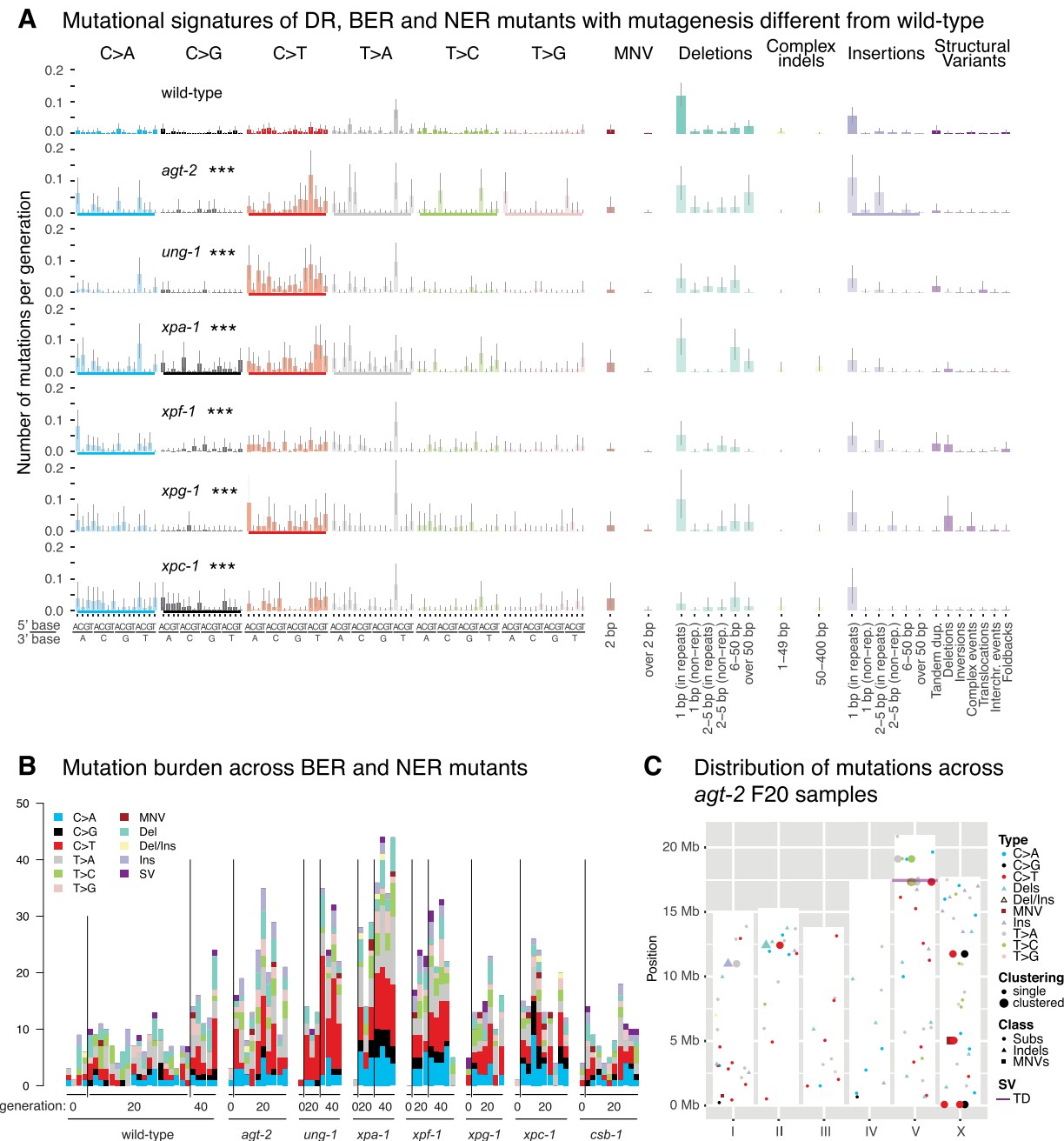

**Fig 3. Mutation analysis of *C. elegans* BER, NER, and DR mutants with mutation rates or spectra different from wild-type. A.** Mutational signatures of BER, NER, and DR mutants that display statistically significantly different mutation spectra than wild-type shown as the number of mutations per generation across all mutation classes. Underscores (bold coloured bars) below each mutation profile indicate mutation types where the total mutation numbers are different from wild-type, three stars indicate genotypes with significantly different rates of substitutions, indels or SVs compared to those in wild-type (FDR < 5%). Single nucleotide variants are shown in the context of their 5' and 3' base context. **B.** Number of mutations of all classes shown for each individual sequenced line of the indicated genotype and generation. The four sequenced wild-type P0 lines reflect the variance present in initial generations. Mutations are shown cumulatively with mutations present in generation F20 included in F40. **C.** Mutation types of all classes and their location on the 6 *C. elegans* chromosomes (I-V and X) observed across *agt-2* mutant lines. The height of the white bars corresponds to the length of each individual chromosome. Single nucleotide variants (SNVs) are indicated by a dot, dinucleotide variants (DNVs) by a square, indels divided in deletions (D), insertions (I), and deletions with insertions (DI) by a triangle, and structural variants (SVs) by a line. Clustered mutations that are present within a single *agt-2* line are depicted by enlarged bold symbols. An analysis of *brca-1*, *him-6* and *smc-6* swas previously shown in ([10] Fig 1C). Information related to the 528 whole genome sequencing WGS primary-source datasets (56 deposited in this study, 472 deposited in (Suppl Data 1 and Supple Note 1 of [10] can be found in S1 Table).

End-joining pathways, classical NHEJ (c-NHEJ) and MMEJ, are typically error-prone and join free DNA ends. c-NHEJ largely acts on blunt DNA ends, while MMEJ requires short DNA resection to generate complementary 2–20 base single-stranded DNA termini to join broken DNA ends [16].

Inactivation of the core components of NHEJ, *cku-80* and *lig-4*, did not produce changes in mutagenesis (Fig 2, S4A Fig). To study the effect of defective HR, we investigated mutation accumulation in *brc-1 brd-1* double mutants. BRC-1/Brca1 and BRD-1/Bard1 proteins form a heterodimer and the corresponding deficiencies are considered epistatic [22]. We report on the double mutant as our genome sequencing analysis revealed that the *brc-1* mutant strain we used also contained a *brd-1* deletion (S1 Table). In contrast to end-joining mutants, the *brc-1 brd-1* double mutant showed increased numbers of single nucleotide variants (Fig 2), small deletions between 5 and 50 bases (Fig 4A), and tandem duplications (TDs) between 1.6 and 500 kbps, with a median of ~12 kbps (Fig 4B and 4C, S5 Fig). Overall, the mutational signature of *C. elegans* BRC-1 BRD-1 deficiency agrees with the flat profile of increased base substitutions described in HR deficient human cancers [23, 24], BRCA1 negative human lymphoblastic MA lines [25] and also resembles the pattern of SVs associated with BRCA1 loss in breast and ovarian cancers [26–28]. HR has been shown to be the predominant DNA repair pathway in *C. elegans* germ cells while NHEJ has a role in somatic cells [29, 30]. Our finding that mutation rates are not elevated in NHEJ mutants is consistent with these observations.

Homologous recombination repair requires substantial end processing at the site of DSBs, which is performed by a series of nucleases creating single-stranded DNA overhangs. As mutants of *C. elegans* nucleases *rad-51*, *mre-11*, and *com-1* are sterile due to defects in meiotic recombination thus precluding MA experiments, we analysed mutation rates in strains deficient for the *rad-51* paralog *rfs-1* and for *rip-1*, which encodes a RFS-1 interacting protein [31]. The RFS-1 RIP-1 complex is thought to stimulate the remodelling of presynaptic RAD-51-coated DNA filaments to facilitate strand invasion for recombinational repair [31]. We observed an overall 2-fold elevated mutagenesis in *rfs-1* and *rip-1* mutants (Fig 2), defined by increased numbers of base substitutions in both strains, increased numbers of small deletions in *rfs-1*, and increased numbers of SVs in *rip-1* (Fig 4A and 4B, S6 Fig). Intriguingly, we observed three 'translocation type' events in *rip-1* but not in any other MA line we analysed (Fig 4B, S6 and S7 Figs). We deduced that these events involved templated insertions of 200–4000 bp sequences, which showed strong homology to multiple genomic regions, including to one homeologous region located as far as 275 kb away from the donor sequence on the same chromosome, accompanied by a deletion of several hundred basepairs at the acceptor site (S7 Fig). These templated insertions may be explained by strand invasion into homeologous template DNA, in line with the pro-recombinogenic role of RIP-1 in mediating RAD-51 dissociation from invading strands [31].

SMC-5 and SMC-6 are components of a ring shaped cohesin complex, considered to tether broken DNA strands to the repair template on the sister chromatid to facilitate HR [32]. *smc-5* and *smc-6* mutants, which have been shown to exhibit defects in meiotic recombination between sister chromatids [32, 33] showed an increased rate of base substitutions and SVs, largely comprised of deletions, tandem duplications, and complex rearrangements (Figs 2, 4A and 4B, S6 Fig). In agreement with a high preponderance of SVs, these lines could only be propagated for up to 5 generations before succumbing to sterility. In contrast to *brc-1 brd-1* mutants, *smc-5* and *smc-6* mutants exhibited a high proportion of large ~10 kb deletions (Fig 4C). Given the role of the ring-shaped SMC-5-6 complex in enforcing close proximity of damaged and template DNA, we hypothesize that double-strand break induced HR is initiated in the absence of the SMC-5-6 complex, but the displacement loop (D-loop) formed following strand invasion falls apart prematurely resulting in the loss of genetic material.

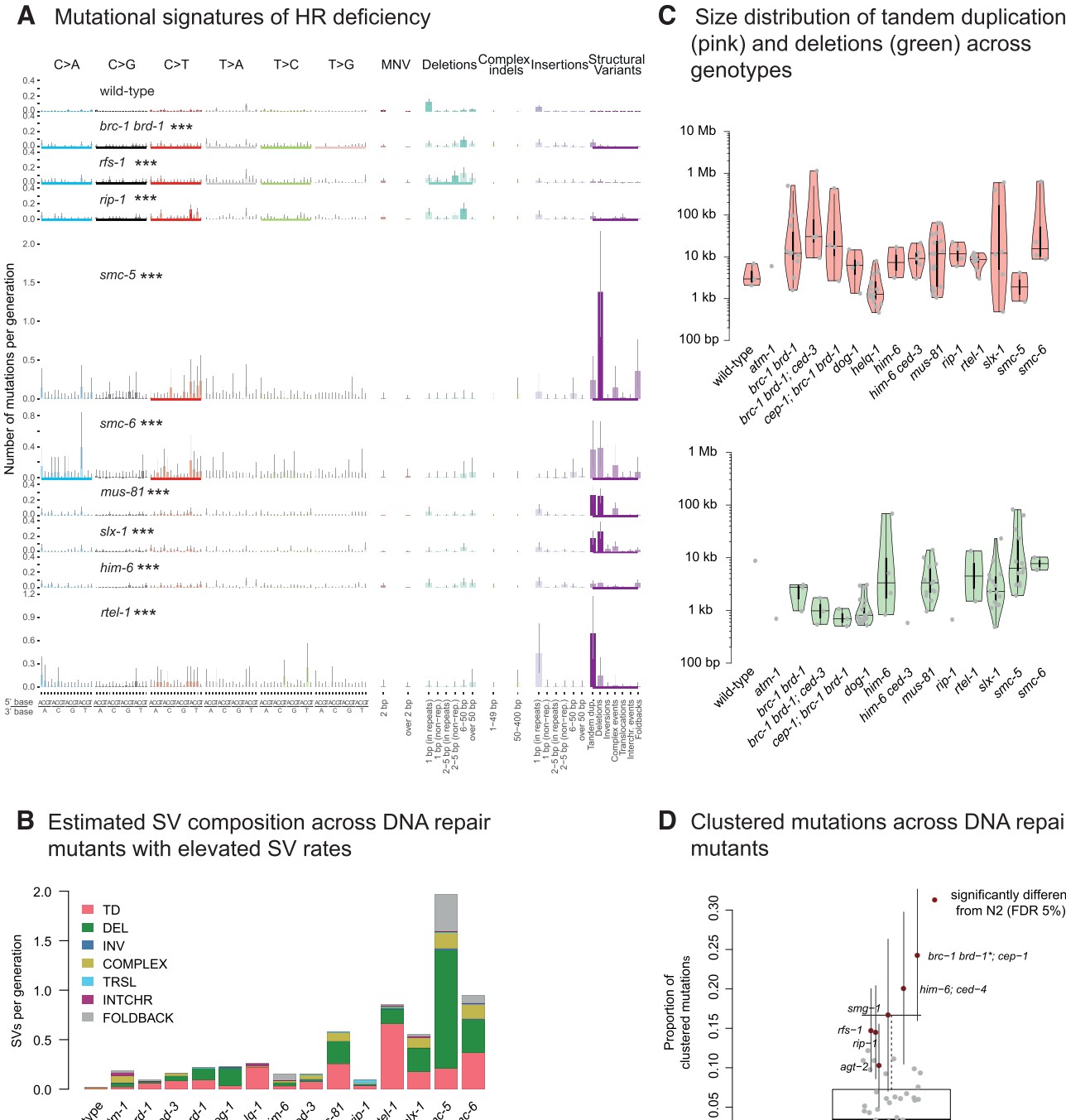

**Fig 4. Mutational signatures and genomic features of mutations in DSBR deficient *C. elegans*. A.** Mutational signatures of DSBR mutants that exhibited statistically significant different mutation rates to wild-type displayed in numbers of mutations per generation. Bold coloured bars denote individual mutation classes where the number of mutations is different from wild-type, an underscore below each mutation profile indicates mutation types with total mutation numbers different from wild-type, and three stars indicate genotypes which have rates of substitutions, indels or SVs significantly different compared to wild-type (FDR < 5%). **B.** Estimated composition of structural variants per generation as estimated for wild-type and DNA repair mutants with elevated SV rates. **C.** Size distributions of tandem duplications (top, pink) and deletions (bottom, green) across wild-type and mutants with elevated SV rates. **D.** Clustering of mutations in DNA repair deficient mutants. Grey dots reflect the average proportions of clustered mutations. Error bars denote 95% confidence intervals. Mutants with a significantly different propensity for mutation clustering from wild-type (dotted black line) are shown and highlighted in red. 'Information related to the 528 whole genome sequencing WGS primary-source datasets (56 deposited in this study, 472 deposited in (Suppl Data 1 and Supple Note 1 of [10] can be found in S1 Table).

The structure-specific nucleases MUS81 and SLX1 act in conjunction to process Holliday Junctions, key four-way DNA intermediates of HR [34–36]. *slx-1* and *mus-81* mutants displayed similar mutational signatures characterized by increased numbers of base substitutions and SVs, with large deletions and TDs being most prevalent (Figs 2, 4A and 4B, S8 Fig). In contrast, the absence of GEN-1, a canonical Holliday Junction resolvase [37], or LEM-3, the ortholog of mammalian Ankle1, recently implicated in the processing of recombination intermediates that persist beyond anaphase [38], did not yield overt changes in mutation rates or signatures (Fig 2, S4A Fig).

DNA helicases are enzymes that unwind double-stranded DNA. Among their multiple roles in HR, they contribute to the unwinding of D-loop structures, a function especially important when a broken DNA end invades a template strand with imperfect sequence homology, thus preventing recombination with homeologous sequences. To investigate mutation patterns induced by helicase deficiencies, we analysed mutants defective for the three *C. elegans* RecQ helicases: *him-6*—the ortholog of the mammalian Bloom syndrome gene which encodes for a helicase involved in HJ resolution and prevention of crossover recombination [39, 40]; *wrn-1*, the ortholog of Werner's syndrome gene which encodes for a helicase possessing an N-terminal 3'-5' exonuclease domain, and capable of resolving aberrant DNA structures with 3' recessed ends [41, 42]; and *rcq-5*, the ortholog of human RECQ5, which displaces RAD-51 from single-stranded DNA and thus prevents excessive recombination [43]. In addition, we analysed lines deficient for *rtel-1* which encodes a conserved helicase involved in genome stability and telomere maintenance [44]. While *rcq-5* and *wrn-1* mutants did not show increased mutagenesis, *him-6* mutants demonstrated 8-fold elevated SV rates compared to wild-type, with 0.15 SVs per generation (SD = 0.03) (Figs 2, 4A and 4B, S4A and S9 Figs). Even more SVs were observed in *rtel-1* mutants with an estimated rate of 0.8 TDs per generation (SD = 0.2) (Figs 2, 4A and 4B), which spanned on average 8 kbps and were generally smaller than TDs observed in *brc-1 brd-1* mutants (Fig 4C). In addition, *rtel-1* deficiency led to ~2.5 fold increase in base substitutions (Figs 2 and 4A). RTEL-1 has a unique role in preventing heterologous recombination during break-induced repair and in promoting non-crossover products [45, 46]. Interestingly, loss of mammalian RTEL1 yields a high number of large deletions and complex rearrangements as a result of excessive crossover and heterologous recombination [45], unlike our data which showed a more simple, tandem duplication signature (Fig 4, S8 Fig). In our experiments, *C. elegans rtel-1* mutants did not grow beyond F15, and most lines became sterile within 5 generations (F5) (S1 Table), suggesting that the absence of RTEL-1 may lead to accumulation of SVs incompatible with organismal viability.

Investigating the genomic context of structural variants, we did not observe any overt changes in the presence of microhomology at the breakpoints compared to wild-type (S4B and S4C Fig). In addition, we confirmed that SVs across almost all HR deficient genetic backgrounds tended to be associated with repetitive DNA regions (S4D Fig), in line with previous reports [47].

Among the HR mutants, we note that *brc-1 brd-1*, *rfs-1*, *rip-1*, *smc-5*, and *smc-6* display elevated levels of base substitutions. Increased base substitutions were also observed in *BRCA1* defective human lymphoblastic MA lines [25]. Moreover, *smg-1*, *rip-1*, and *rfs-1* exhibit evidence of mutational clustering (Fig 4D), with about 15% of base substitutions occurring within distances smaller than 1 kbps (Materials and Methods). Clusters of mutations may arise through error-prone polymerases reading across lesions [48–50]. In addition, the NHEJ or MMEJ error-prone DSB repair pathways can also generate clustered mutations when DNA strands with incompatible ends are joined together [51]. The absence of clustered mutations in NHEJ or MMEJ mutants could be explained by the action of redundant error-free HR pathways.

## Translesion synthesis (TLS)

TLS polymerases are specialised DNA polymerases that replicate across and insert nucleotides opposite damaged bases. Depending on the inserted nucleotide, this results in error-free or error-prone lesion bypass [52]. *C. elegans rev-3/REV3L* mutants, deficient in the catalytic sub-unit of polymerase ζ, accumulated increased numbers of 50–400 bp deletions (Figs 2 and 5A, S10 Fig). Similarly, *polh-1/POLH (lf31)* and *polh-1/POLH (ok3317)*, DNA polymerase η mutants, displayed 50–400 bp deletions, with only *polh-1(lf31)* reaching clear statistical significance over the generations tested (Fig 2, S10 Fig). Our data suggest that REV-3, and likely POLH-1, prevent DNA breaks by reading across damaged bases that also occur in the absence of exogenous DNA damage. Our results on REV-3 and POLH-1 are in line with previous findings in *C. elegans* [18, 53–55], yeast, and mammalian cells [56, 57].

In addition, we observed a slightly increased base substitutions rate, namely for C>T changes, in *polk-1* (Fig 5A, S10 Fig), which may indicate a role in error-free bypass of endogenously arising guanine modifications [58]. Mutants defective in REV-1 TLS polymerase did not demonstrate a significant and reliable change in mutagenesis compared to wild-type (S10 Fig).

## Mutation accumulation in mutants deficient for DNA crosslink repair

The repair of DNA interstrand crosslinks (ICL) provides a formidable task. It involves the Fanconi Anaemia (FA) proteins required for sensing ICLs and assembling various repair factors at the site of damage [59–61]. Here we investigate mutant lines defective for FNCM-1/FANCM, a helicase involved in DNA damage recognition, FANCI-1/FANCI, and FCD-2/FANCD2, the key Fanconi repair proteins ubiquitinated by an E3 ubiquitin ligase complex and thought to assemble proteins required for ICL processing. We did not observe overt differences in mutagenesis between *fcd-2, fncm-1*, or *fnci-1* mutants, and wild-type, suggesting that the *C. elegans* Fanconi Anemia ICL repair pathway does not significantly contribute to the repair of DNA damage that occurs under normal, unchallenged growth conditions (Fig 2, S11A Fig). FAN-1/FAN1 is a conserved structure-specific DNA nuclease that can resolve ICLs independently of the FA pathway [62–64]. As for the core FA components, we did not observe elevated mutation rates in *fan-1* mutant lines (Fig 2, S11A Fig).

DOG-1, the *C. elegans* ortholog of the mammalian FANCJ helicase, facilitates error-free replication through DNA tertiary structures formed by G-rich DNA sequences, referred to as G-quadruplexes [65–67]. *dog-1* mutants exhibited increased mutagenesis (Fig 2) as previously described [67], with 6-fold higher numbers of 50–400 base pair indels and 13 fold more SVs, predominantly deletions (Fig 5A, S9 Fig). Across all 11 *dog-1* deficient samples, 81% of long deletions (17/21) and 78% (109/139) of shorter, 50–400 bp deletions overlapped with one of the 4291 regions in the *C. elegans* genome predicted to form G-quadruplex structures [68], in line with previous reports [67] (Fig 5B, S9 and S11B Figs). We found that G-quadruplex induced deletions occurred at a frequency of about 1 lesion per generation in *dog-1* mutants. We rarely observed SVs associated with G-quadruplex forming sequences in other DNA repair mutants (Fig 5B, S5, S6, S8, S9 Figs), including *him-6*, encoding the *C. elegans* ortholog of the mammalian BLM helicase (S9 Fig), which has been shown to prevent replication fork stalling at G-quadruplex sites in human and murine cells in conjunction with FANCJ [69].

Another helicase mutant that displayed a distinct phenotype was *helq-1*/HELQ, encoding for a conserved helicase and thought to act in DNA crosslink repair in a pathway separate from FCD-2 [70]. In addition, *helq-1* has been shown to be synthetically lethal with *rfs-1*, due to its role in resolving DSB repair intermediates during meiosis [71]. *helq-1* deficient lines showed an increased proportion of tandem duplications (TDs) compared to other strains

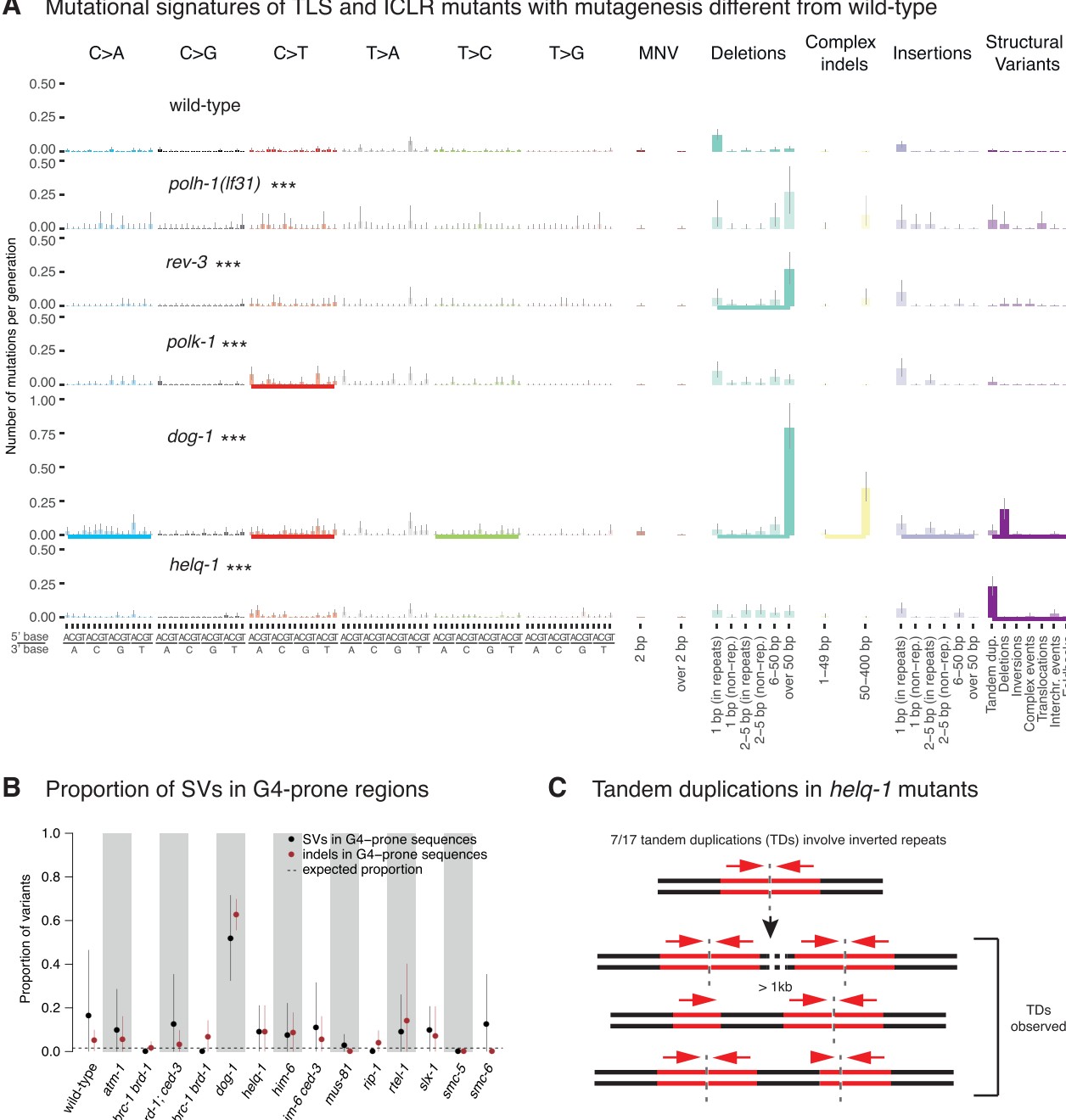

**Fig 5. Signatures and genomic features of mutations in TLS and ICLR deficient *C. elegans*. A.** Mutational signatures of TLS and ICLR mutants that exhibited statistically significant differences to wild-type mutation rates displayed in numbers of mutations per generation. Same layout as Fig 4A. **B**. Proportion of indels (brown) and SVs (black) in G-rich regions in wild-type and across genotypes with elevated rates of SVs. Dotted line represents the proportion of variants falling into these regions as expected by chance. **C.** Tandem duplications (TDs) in *helq-1* mutants. An analysis of *rev-3* was previously shown in ([10] Fig 1C). Information related to the 528 whole genome sequencing WGS primary-source datasets (56 deposited in this study, 472 deposited in (Suppl Data 1 and Supple Note 1 of [10] can be found in S1 Table).

(Figs 4B and 5A, S5 Fig). TDs ranged in size between 457 and 8089 bp with a median of 1270 bp (Fig 4C). The mutational spectrum of *helq-1* differed from that of *brc-1* mutants, in which deletions of 6–50 bp and TD (with a median size of ~12 kbps) were observed with comparable frequency (Fig 4B and 4C, S5 Fig). Interestingly, 41% (7/17) of TDs in *helq-1* mutants were associated with inverted repeat sequences (S2 Table). To investigate how TDs present at inverted repeats relate to DNA replication directionality, we used the closest origin of replication as a reference point and tested the correlation between the orientation of the TD breakpoints and the direction of leading strand synthesis. Out of the 7 inverted repeat-associated TDs, we could determine the directionality of replication in 5 cases (Materials and Methods, S2 Table). In 4 cases, TD oriented in line with leading strand replication. In 3 of these, inverted repeats were present downstream of the TD, and in one case upstream (Materials and Methods, S2 Table). We speculate how these tandem duplications may arise in the discussion. In summary, our data suggest that the DNA helicases HELQ-1 and DOG-1 are required to facilitate replication fork passage through distinct secondary structures (Fig 5B and 5C). While DOG-1 is needed for the passage through G-rich structures [65], HELQ-1 may help to overcome stem loop structures, possibly on the lagging strand.

## Mutations and subtelomeric chromosome fusions in ATM-1 defective strains

ATM is a conserved PI3 kinase involved in DNA damage checkpoint regulation and telomere homeostasis. ATM deficiency has been reported to be associated with shorter telomeres, from yeast to mammalian cells. Moreover, in ATM deficient yeasts, *C. elegans* and *Drosophila*, the last depending on retrotransposon transposition rather than telomerase activity for telomere maintenance, chromosome fusions have been observed cytologically or through sequencing of PCR products across chromosomes [72–77]. *C. elegans atm-1/ATM* mutants are hypersensitive to ionizing radiation (IR) [78, 79]. In addition, *atm-1* lines propagated over multiple generations have been described to display a stochastic *him* (high incidence of males) phenotype, an indicator of meiotic chromosome mis-segregation associated with sex chromosome to autosome fusions [78, 80]. Investigating mutation rates in *atm-1* lines grown for 20 generations, we observed 2-fold elevated numbers of SVs, predominantly inversions (Figs 2 and 6A). This increased incidence of SVs agrees with previous estimates of mutation rates in *atm-1* mutants which were based on scoring the number of essential mutations in *atm-1* backgrounds [78]. Interestingly, 4 of the 5 *atm-1* lines grown for 20 generations carried SVs (with over 70% (9/11) of all observed SVs) localised in subtelomeric regions (Fig 6B, S5 Fig), unlike any other DNA repair deficient mutants examined. Most subtelomeric SVs (8/9) could be classified as complex rearrangements with at least 2 overlapping or adjacent events, often associated with copy number changes (Fig 6C, S5 and S13 Figs). Moreover, 4/5 complex rearrangements involving a single chromosome end displayed a loss of telomere sequences, suggesting a possible end-to-end fusion between homologous chromosomes. Furthermore, we observed 2 cases of interchromosomal rearrangements between different autosomes with breakpoints in subtelomeric regions associated with deletion of telomere sequences and copy number alterations (Fig 6B, S5 and S13 Figs). None of the *atm-1* lines carried translocations or amplifications of the genomic regions termed TALT1 or TALT2, amplified in survivors of telomerase-deficient *C. elegans* strains, and considered to be utilized as templates for an alternative (telomerase independent) telomere lengthening (ALT) mechanism [81]. Similarly, we did not observe translocation events associated with *atm-1* SVs (S5 and S13 Figs) making templated telomere maintenance from interstitial telomere sequences buried in the genome unlikely.

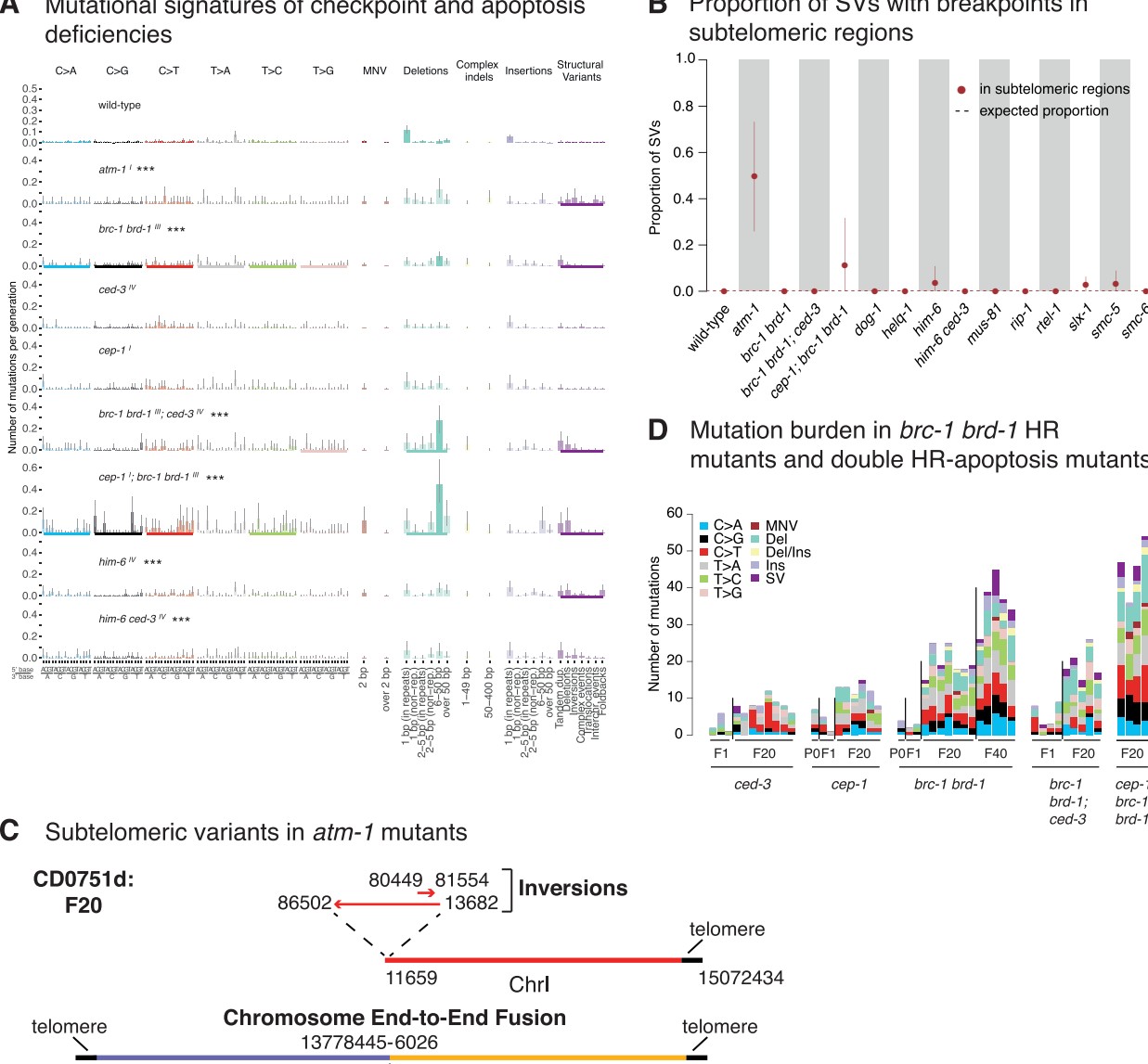

**Fig 6. Signatures and genomic features of mutations in DNA damage checkpoint and apoptosis deficient *C. elegans*. A.** Mutational signatures of mutants that exhibited statistically significant differences to wild-type mutation rates. The chromosomes on which respective genes are located are indicated in superscript following each mutant name. Layout as Fig 4A. **B.** Proportion of SVs with breakpoints into subtelomeric regions across wild-type and mutants that exhibit elevated SV rates. Dotted lines represent the fraction of variants expected to occur in subtelomeric regions by chance. **C.** Examples of subtelomeric structural variants in *atm-1* mutants. **D.** Quantification of mutation burden in indicated DNA repair mutants for initial generations and F20 and F40 generations as shown. Information related to the 528 whole genome sequencing WGS primary-source datasets (56 deposited in this study, 472 deposited in (Suppl Data 1 and Supple Note 1 of [10] can be found in S1 Table).

We speculate that loss of *C. elegans atm-1* function could lead to reduced recruitment or access of telomerase to the shortest telomeres consistent with studies in tel1/ATM mutant yeast [82–84], thereby leading to telomere loss and chromosome fusions. Access of telomerase to its telomeric substrate could be inhibited by *atm-1* loss due to reduced end resection, a mechanism also discussed for *atm*$^{-/-}$ dependent telomere shortening and fusions in mammalian cells [85].

**p53 and apoptosis defective strains.** In the presence of excessive DNA damage, cells can trigger the p53 pathway to activate their apoptotic demise. CED-3 and CED-4, orthologs of a human caspase and the APAF1 protein, are essential for DNA damage induced and developmental apoptosis [86]. In contrast, CEP-1, the *C. elegans* p53 ortholog is specifically required for DNA damage induced apoptosis [87, 88]. We did not observe increased mutagenesis in *ced-3*, *ced-4*, or *cep-1* defective mutants, indicating no major role of DNA damage induced apoptosis in preventing mutagenesis under unchallenged growth conditions (S14 Fig). Our dataset also includes deletions of spindle assembly checkpoint (SAC) genes. The spindle assembly checkpoint delays anaphase progression until all chromosomes are correctly attached to the mitotic spindle apparatus, thereby ensuring faithful chromosome segregation [89]. In addition, SAC has been implicated in DSB repair consistent with *bub-3* and *san-1* SAC mutants exhibiting IR sensitivity [90]. Lines deficient for *bub-3* and *san-1*, corresponding to human BUB3 and BUB1B proteins, respectively [91], did not show increased mutagenesis (Fig 2, S14 Fig).

Having observed distinct mutational patterns in HR deficient mutants, we wanted to test whether combining HR mutants with apoptosis and/or DNA damage response deficiency would lead to increased or altered mutagenesis. Increased *cep-1/p53* dependent germ cell apoptosis has been reported in a number of HR mutants such as *him-6* and *brc-1*, suggesting that a higher number of nuclei carry increased or unrepaired DNA damage which might be eliminated by *cep-1* dependent apoptosis [22, 40, 92, 93]. Double mutants of *him-6* with *ced-3* (6 lines), *ced-4* (3 lines) or *cep-1* (3 lines) did not display changes in mutation rates or spectra compared to *him-6* single mutants apart from 2 out of 3 *him-6; ced-4* lines exhibiting a variable amount of clustering (S14 Fig, Fig 4D). In contrast, 4 *cep-1; brc-1 brd-1* lines (but not the 5 *brc-1 brd-1; ced-3* lines*)* showed an increased rate of mutagenesis compared to *brc-1 brd-1* alone (Fig 6A and 6D). Specifically, *cep-1* inactivation exaggerated the mutational features of the *brc-1 brd-1* mutant, leading to increased incidence of small deletions and structural variants (Fig 6A). We also observed prominent clustering of base substitutions in all triple mutant lines averaging to over 20% of mutations being clustered (Fig 4D). Thus, at least for HR deficiency conferred by *brc-1 brd-1*, additional *cep-1* inactivation increases mutation clustering. Given that apoptosis deficiency in *brc-1 brd-1; ced-3* lines does not result in increased mutagenesis, we speculate that increased mutagenesis in *brc-1 brd-1; cep-1* lines might be associated with a role of CEP-1, independent of apoptosis regulation. CEP-1 could either trigger the cell-cycle checkpoint or facilitate more efficient DNA repair.

## Discussion

Here, extending on our previous study primarily focused on the effects of genotoxic agents [10], we systematically catalogued the mutational characteristics of DNA repair deficiencies across all conserved *C. elegans* DNA repair and damage response pathways in inbred lines propagated for up to 40 generations. Our data provide a comprehensive picture of the contributions of various DNA repair and damage response pathways towards genome integrity in an experimental system under spontaneous, endogenous conditions not challenged by mutagen exposure. Except for mismatch repair deficiency [11, 47], defined repair deficiencies only lead to modest effects, with a 2–5 fold increase of mutations in 44% of all strains tested, with notable examples in almost every pathway. Alkylguanine alkyltransferase, Uracil glycosylase and NER deficiency are associated with increased acquisition of base changes. TLS mutants *rev-3*(pol $\zeta$) and *polh-1*(pol η) show elevated numbers of 50–400 bps deletions. Interestingly, HR deficiency can manifest in different ways. *brc-1/Brca1* and *rad-51* paralog mutants show elevated mutagenesis across most types of mutations. Other HR mutants inactivating the MUS-81 and

SLX-1 nucleases, and the HIM-6/BLM, HELQ-1 and RTEL-1 helicases are associated with increased numbers of SVs. DOG-1 has a unique role in preventing deletions next to G-rich sequences, while the HELQ helicase may contribute to faithful replication across secondary DNA structures such as inverted repeats. RIP-1 appears to avert templated insertion into homeologous sequences. The ATM-1 checkpoint kinase prevents chromosome end-to-end fusions. Finally, deficiency of the *p53* like gene *cep-1* exacerbates mutagenesis caused by HR defects.

## Redundancy of DNA repair pathways

It is well established that thousands of DNA lesions occur spontaneously during each cell-cycle and that the vast majority of DNA lesions are repaired. Thus, the absence of significant mutagenic effects in the majority of genotypes tested underpins a high level of redundancy within and among different DNA repair pathways. Moreover, it may require the combined deficiency of multiple DNA repair pathways to trigger excessive mutagenesis in the germline. Such reasoning is in line with our recent whole genome analyses showing that multiple pathways act in concert to repair DNA lesions caused by the exposure to known genotoxins, with deficiencies of different pathways potentially leading to increased mutagenesis and/or altered mutagenic signatures [10]. Equally, latent defects such as those caused by the deletion of the non-essential polymerase subunit *pole-4*, only become apparent in conjunction with a DNA repair deficiency [11]. While *pole-4* alone does not cause increased mutagenesis, combining this mutant with MMR causes mutagenesis beyond what is observed upon MMR alone [11]. Many cases of DNA repair pathway redundancies have been described, for instance, simultaneously defective TLS, NER, and MMEJ renders *C. elegans* sensitive to normal levels of daylight [94].

In contrast to a recent large-scale mutation accumulation screen in budding yeast [47], we did not observe widespread copy number changes in our analysis, likely because most such changes are incompatible with viability and fertility of a multicellular organism such as *C. elegans*. It is possible that nematodes suffering from gross chromosomal alterations are lost during propagation across generations. However, using our experimental set-up, we were previously able to detect instances of severe chromosomal rearrangements, including complex chromosome fusion events that contain scars indicative of breakage-fusion-bridge cycles and chromothripsis [12]. Nevertheless, it is likely that we underestimate mutation rates, especially in strains that could not be propagated for 40 generations, namely those defective for the RTEL-1 helicase and the SMC-5 and SMC-6 cohesion proteins.

## Mechanistic insights

Our detailed characterisation of mutation rates, mutational signatures and localised mutation features, combined with the known enzymology of many repair enzymes, provides mechanistic insights: Our data confirm the specific role of the DOG-1/FANCJ helicase in unwinding G-quadruplex forming sequences [95], and we show that this feature is unique among the helicase mutants we analysed. We also reveal a specific role of the HELQ-1 helicase frequently in the context of repetitive sequences such as inverted repeats. At present, we can only speculate how these tandem duplications might arise. It is likely that the genesis of TDs involves microhomology-mediated break-induced replication. Inverted repeats may form secondary stem-loop structures that impede replication fork progression (Fig 5C, S12 Fig). In the case of the presence of a stem-loop structure in the DNA ahead of a replication fork, stalling could occur during the attempt to unwind the secondary DNA structure. The replication machinery may re-initiate at a downstream template, resulting in a duplicated region, before successfully replicating through the stem loop structure in a second attempt. In such a scenario, tandem duplications would always occur upstream of the inverted repeat (Fig 5C, S2 Table). Alternatively,

an inverted repeat could more readily adopt a stem loop structure in the extended single-stranded region of the lagging strand, which is particularly prone to form secondary structures (Fig 5C, S12 Fig). Such stem loop structures would be prime substrates for HELQ, which has been shown to bind to single-stranded DNA and act as a 3' to 5' helicase, thus facilitating the dissolution of the stem loop [96]. In the absence of HELQ activity, these stem loops could be recognised by a nuclease (S12 Fig) and a resulting single-strand break might facilitate the invasion into the freshly replicated leading strand DNA and prime break-induced replication. Recapturing the original template (S12 Fig, step 3) after break-induced replication would restore the fork, resulting in a tandem duplication in only one of the two chromatids (S12 Fig). Crucially, the position of the nucleolytic cut, upstream, downstream or within the inverted repeat, and the position at which invasion into the template strand occurs, would determine the breakpoint of the tandem duplication.

In summary, our data suggest that the DNA helicases HELQ-1 and DOG-1 are required to facilitate replication fork passage through distinct secondary structures (Fig 5B and 5C). While DOG-1 is needed for the passage through G-rich structures [65], HELQ-1 may help to overcome stem loop structures, possibly on the lagging strand.

Furthermore, cases of unique gene conversion events into homeologous sequences in *rip-1* mutants support a role of RAD-51 paralogs in preventing homeologous recombination. This is in line with biochemical activity of the RFS-1 RIP-1 paralog complex in remodelling presynaptic RAD-51-containing DNA filaments to facilitate strand invasion for recombinational repair [31]. Finally, we provide evidence that the ATM-1 checkpoint kinase has a specific role in protecting sub-telomeric repeats from DSBs and preventing deletions, inversions and chromosome fusions.

Based on their mutational signatures, strains defective for homologous recombination can be broadly grouped into two classes. First, BRC-1 and RAD-51 paralog mutants show elevated numbers of point mutations, as well as increased numbers of small deletions, and structural variants. A similar pattern was observed in a study of HR knockouts in chicken DT40 cell lines [25]. We suspect that increased point mutations might be a scar indicative of error prone translesion synthesis, necessary when damaged bases are neither repaired by BER and NER nor by replication fork reversal which is linked to recombinational repair [97]. Point mutations and small deletions also occur when HR is replaced by more error-prone NHEJ or MMEJ pathways, the latter being associated with the occurrence of small deletions in human BRCA1 mutants [28]. Conversely, deficiencies of other HR proteins, like SLX-1 and MUS-81, and helicases including HIM-6, RTEL-1 and HELQ-1, are associated with a specific increase of SVs. We speculate that these proteins may not have a role in HR pathways directly linked with DNA replication (see below).

## Nature of germ cell divisions and mutagenesis

It is important to keep in mind that germ cell mutagenesis might occur at different stages of the *C. elegans* life cycle. The nature of germ cell divisions is fundamentally different across various developmental stages. During the invariant embryonic development of *C. elegans*, the germ cell lineage is defined by 3 asymmetric cell divisions, and from the first zygotic cell division onwards, a single posterior daughter cell always defines the germ line, which finally splits as a part of the 4[th] germ cell division into the two founder cells [98]. Starting from the L1 larval stage, each of these founders within a timeframe of three days expands to form one of the two gonads comprising ~1000 germ cells [98]. Embryonic germ cell divisions occur very rapidly within a timeframe of less than 20 minutes, and cells are largely refractory to DNA damage checkpoints. Evidence exists that translesion polymerases are particularly important during

this stage [99, 100]. The first cell-cycle in developing germ lines occurs after an extended period of transcriptional quiescence, and synchronized transcriptional onset appears particularly challenging for genome integrity [101, 102], all the more so that global transcriptional induction in these cells required topoisomerase II induced DNA double strand breakage which has to be mended [101]. In contrast, germ cells residing in the adult germ line are subject to cell-cycle and apoptosis checkpoints, and take an excess of 10 hours to complete [86]. It appears possible that the increased number of mutations observed in *cep-1 brc-1* double mutants reflects a role of the CEP-1 p53 like protein in preventing excessive mutagenesis. CEP-1 is expressed in mitotically dividing germ cells, as well as in late pachytene where late stages of meiotic recombination occur. Given that we did not observe comparable mutagenesis in *brc-1* and apoptosis defective double mutants, apoptosis being restricted to pachytene cells, we speculate that mutagenesis might reflect a role of CEP-1 in mitotically dividing germ cells, possibly affecting the cell-cycle checkpoint or DNA damage response. Finally, it is reasonable to suggest that many lesions we observe to accumulate in our transgenerational set-up occur in germ cells, especially during meiosis. Indeed, it appears plausible that many SVs might be associated with meiotic recombination. A large excess of DSBs are generated by the SPO-11 nuclease during meiosis, and typically only one DSB per chromosome pair matures into a crossover to facilitate the exchange of genetic information between maternal and paternal chromosomes [4]. Many SVs we observed in HR mutants, particularly those that did not demonstrate an excessive accumulation of point mutations, likely result from faulty recombinational events during meiosis: For instance, the SLX-1 and MUS-81 nucleases, and the HIM-6 helicase contribute to the resolution of meiotic Holliday junction intermediates [34, 103, 104]. In contrast, BRC-1 and the SMC-5/6 complex are implicated in the repair of the excess meiotic DSBs not engaged in crossover recombination [33].All in all, we provide a comprehensive view of how the DNA repair and damage response machinery acts to preserve genome integrity over generations. Even in the absence of strong exogeneous genotoxins, multiple DNA repair pathways are required to mend endogenous DNA damage. In wild-type, background mutagenesis leads to a relatively uniform mutation pattern. A similar flat mutational signature has been observed in the human germline and in somatic tissues [6, 9]. It is tempting to assume that a diverse set of DNA repair processes is also constantly operating in human germ and somatic cells during normal proliferation to ensure a highest possible level of genomic integrity, resulting in a flat residual mutational signature. It will be interesting to extend our studies to human inherited conditions, where defective DNA repair is associated with progeria, developmental defects, microcephaly, spinocerebellar ataxia, and cancer. Studying both experimental human cell models and patient samples will allow testing if increased mutagenesis occurs and if certain mutational features correlate with disease phenotypes.

## Materials and methods

### *C. elegans* strains, propagation and maintenance

All *C. elegans* strains used in this study, newly and previously generated [10–12] (S1 Table) were backcrossed 6 times against the wild-type N2 Bristol reference strain TG1813 [10–12], (Fig 1A). The majority of strains were clonally propagated for 20 or 40 generations as described previously [10]. *rtel-1*, *smc-5*, *smc-6* lines were grown for 5 generations as these lines tended to become sterile when grown for more generations (S1 Table for number of lines and generations per genotype). As described in detail [10] 5–10 single L4 stage hermaphrodites (F1s) were randomly chosen for each genotype and transferred to individual 1× NGM plates seeded with OP50 bacteria. Every 3–4 days, 1 single L4 hermaphrodite was randomly chosen among the progeny per plate and individually propagated further, a process repeated until the

indicated generation (F5, F20 and F40). Once final generation hermaphrodites had produced clonal progeny 5 lines were transferred to 9 cm 3× NGM plates and allowed to reach starvation. Mixed stage worms (the majority of which two generations the final transferred L4) were washed off plates, washed 3× in M9 medium, pelleted and frozen in liquid nitrogen [10]. Genomic DNA was isolated from three samples using the Invitrogen ChargeSwitch® gDNA Mini Tissue Kit (Thermo Fisher Scientific, CS11204) [12].

## Variant analysis

The relevant procedures are described in detail [10]. In summary, DNA sequencing was performed using Illumina HiSeq 2000 and 10X Genomics short reads sequencing platforms at 100 bp paired-end, with a mean coverage of 50x. The resulting reads were then put through Sanger Cancer IT Pipeline, including alignment with BWA [105–108] against WBcel235.74. dna.toplevel.fa as the reference genome (http://ftp.ensembl.org/pub/release-74/fasta/ caenorhabditis_elegans/dna/), CaVEMan for base substitution calling [105–107], Pindel for indel calling [105, 106]. Structural variants were called manually using DELLY [105] for calling structural variants and deletions/duplications longer than 400 bps. All variant calling procedures for each sample used a dedicated P0 sample from the same genotype group, or one wild-type P0 sample as a control (S1 Table).

Resulting variants were filtered based on the site coverage, number and orientation of the reads, supporting the variant in the test and control samples, overlap with other variants (relevant for substitutions and indels in homopolymer tracks), and a panel of 6 wild-type samples (for more detailed filtering description, see [10, 11]). In addition, we filtered out the recurrent variants between unrelated samples to ensure the absence of technical artifacts. The variants in samples with generation higher than 1 were filtered against all the P0 and F1 line samples of the same genotype. Mutations were classified based on their size and context: base substitutions were classified into single base substitutions, further split into 96 types by mutation type (C>A, C>G, C>T, T>A, T>C, T>G), and trinucleotide context based on pyrimidine reference, di- and multi-nucleotide variants; indels were classified based on event type (deletion, insertion, or complex indel), local context (whether it falls into a repetitive region, only for events smaller than 5 bp), and size (1bp, 2–5 bp, 5–50 bp, 50–400 bp); and structural variants were classified into deletions, tandem duplications, inversions, intra- and interchromosomal translocations, foldbacks or complex events based on the breakpoint orientation, and proximity (for more details, see [10]). Sample information and their corresponding ENA accession codes are listed in S1 Table. Filtered variant sets for the samples already published [10] are available in the supplementary data of the respective publication. Filtered variants of a further 31 samples analysed in this study are provided in S1 File.

## Mutation rates and mutational signatures calculations

Total mutation rates, as well as the rates of base substitutions, indels and structural variants, for each genotype were expressed in mutations acquired per generation and were estimated using additive non-negative Poisson regression using the samples with generation higher than 1. Every sample $i$, $i = 1,\ldots N$ out of $N = 528$ was assigned a number of mutations of the category of interest, $Y_i \in \mathbb{N} \cup \{0\}$. For a vector of mutation counts across all samples $Y$, $Y = \{Y_i\}_{i=1}^N$, we calculated the mutation rates per generation for each genotype using the following model:

$$Y \sim \text{Poisson}(\lambda), \lambda = g \cdot G \cdot \mu,$$

where $g \in R_+^N$ is the adjusted number of generations which takes into account the 25% chances of a heterozygous mutation to be lost or to become fixed [12], $G \in Mat_{N \times K}(\{0,1\})$ is a binary

matrix indicating the genotype of each sample, and $\mu \in R_+^K$ is a non-negative vector of mutation rates per genotype, with $K = 62$ being the number of genotypes analysed.

To calculate the mutational signatures, which consisted of mutation rates across all $R = 119$ mutation types (96 single base substitutions, 2 types of multi-nucleotide substitutions, 14 types of indels and 7 types of structural variants), we used a negative binomial model to account for a higher variance compared to the mean in individual counts and a high amount of zero values. For a matrix of counts $\underline{Y} \in Mat_{N \times R}(N \cup \{0\})$, the matrix of signatures $S \in Mat_{R \times K}(R_+ \cup \{0\})$ was calculated using the following model (where $S^T$ denotes a transposed matrix $S$):

$$\underline{Y} \sim \text{NegativeBinomial}(\chi, \phi), \chi = g \cdot G \cdot S^T, \phi = 100.$$

The parameter $\phi = 100$ was chosen empirically based on the estimations from the data, and suggests a slight deviation from Poisson model towards higher variance. Signatures were estimated using the log-normal prior $S_{ij} \sim logN(0, \sigma^2)$ with a fixed parameter $\sigma^2$ estimated from the data.

The estimates of the posterior means for total mutation rates in each genotype, as well as the mutation rates per type/class, were obtained using Hamiltonian Monte Carlo sampling procedure with at least 1000 warm-up and 1000 post warm-up samples [109]. As these estimates were assumed to be log-normally distributed, mutation rates in DNA repair deficient genotypes were compared to the respective entity for wild-type by testing whether a difference between their logarithms followed a normal distribution, or, equivalently, if their squared log ratio followed a chi-squared distribution. Resulting two-sided p-values were corrected for multiple testing across all genotypes using the Benjamini-Hochberg procedure [110]. Mutation rates per base pair per cell division were calculated assuming 15 cell divisions per generation and 2 copies of the 100,272,607 bp long nuclear genome.

### Analysis of repetitive regions and G4-prone sequences

Genome-wide G4-prone sequences for *C. elegans* [68], and repetitive regions as deposited in Repbase (www.girinst.org/downloads/repeatmaps/C.Elegans) were used to determine the association of SVs with specific genomic regions [111]. For each SV, 60 bp regions around the breakpoints were overlapped with the location of G4-prone and repetitive regions. Only unique variants were used to calculate the associated proportions for each genotype. Proportions expected by chance were estimated as the ratios between the sums of all regions of interest, and *C. elegans* genome size.

### Relationship to replication directionality

Directionality of replication was determined using Okazaki fragment sequencing from [112] performed on different developmental stages of *C. elegans*. To identify the directions of replication across the genome, we split the genome in 100 bp bins and calculated the fractions of Okazaki fragment reads on the minus strand within each bin for each of the 6 samples analysed in the study:

$$t_j^l = \frac{t_{j-}^l - t_{j+}^l}{t_{j-}^l + t_{j+}^l},$$

where $j = 1\ldots,6$ denotes the sample index, and $l$ denotes the index of the bin. The bins where the average across samples was greater than its standard deviation, i.e. $|mean_j(t_j^l)| > 2 \cdot sd_j(t_j^l)$, were assigned a "+" direction (or called right-replicating) if $mean_j(t_j^l) > 0$, and a "-" direction (or called left-replicating regions) if $mean_j(t_j^l) < 0$. The bins where the standard deviation of

the fraction of minus strand Okazaki fragment reads was greater than its average ($|mean_j(t_j^l)| < 2 \cdot sd_j(t_j^l)$) were considered as having an ambiguous replication direction. In total, we inferred the direction of replication for 45% of the genome.

## Analysis of clustered mutations

Clustering of mutations was assessed using the start points of all base substitutions and indels across samples of the same genotype and generation. Clustered status was assigned based on a sliding window of 1000 bp. Estimates of the proportion of clustered mutations were obtained from a linear model using samples with generation higher than 1 in genetic backgrounds with >3 such samples: **Proportion_clustered ~ rates** $+ \varepsilon, \varepsilon \sim$ N(0,$\sigma^2$), and all DNA repair deficient backgrounds were compared to that in wild-type by the following Z test: $Z = \frac{r_g - r_{wt}}{\sqrt{SE(r_g)^2 + SE(r_{wt})^2}}$

[113], where $r_g$ denotes the rate for a DNA repair deficient genotype *g*. False discovery rate among the resulting p-values was corrected for multiple testing using Benjamini-Hochberg procedure [110].

## Microhomologies at SV and indel breakpoints

Microhomologies (MH) at the breakpoints of SVs and indels were assessed by measuring the lengths of precise alignments around each breakpoint, calculated as a sum of perfect alignment lengths between the two 30 bp regions upstream, and two 30 bp regions downstream from the breakpoint sites. Only unique SVs and indels were used to calculate the proportions of variants with MH for each genotype.

## Supporting information

**S1 Fig. Summary of mutations acquired in the screen.** Total number of mutations (substitutions in black, indels in green, and structural variants in purple) per sample across wild-type and DNA repair mutant lines. The black line denotes the median number of mutations for each mutation class across all experiments.
(EPS)

**S2 Fig. Mutational signatures and distribution of mutations in DR and BER deficient mutants. A.** Mutational signatures of all tested DR and BER mutants, shown as numbers of mutations per generation. Bold coloured bars below each mutation profile indicate mutation types for which the sum across individual classes is different from wild-type. Three stars indicate genotypes with significantly different rates of substitutions, indels or SVs compared to wild-type with FDR < 5%. **B.** Distribution of mutations across chromosomes for DR and BER deficient samples. Only *agt-2* shows a significant degree of mutation clustering. The pink and orange shaded regions in *parp-1* and *parp-2* samples, respectively, indicate the location and extent of observed structural variants (pink—TD (tandem duplication); orange—Fold (fold-back duplication)).
(EPS)

**S3 Fig. Mutational signatures and distribution of mutations in NER deficient mutants. A.** Mutational signatures of all tested NER mutants displayed as numbers of mutations per generation. Bright bars denote individual mutation classes for which mutation numbers differ significantly from wild-type. Bold coloured lines below each mutation profile indicate a wider range of mutation types for which total mutation numbers are different from wild-type. Three stars indicate genotypes with significantly different rates of substitutions, indels or SVs compared to wild-type (FDR < 5%). **B.** Comparison between the humanised version of the

combined mutational spectrum across all NER deficient *C. elegans* samples and COSMIC signatures SBS5 and SBS8 previously associated with NER deficiency (cosine similarity scores 0.64 and 0.56, respectively). No other COSMIC SBS signatures showed significant similarity to the *C. elegans* NER spectrum.
(EPS)

**S4 Fig. Mutational signatures and characteristics of mutagenesis in DSBR mutants. A.** Mutational signatures of all tested DSBR mutants displayed as numbers of mutations per generation. Bright bars denote individual mutation classes for which mutation numbers differ significantly from wild-type. Bold coloured lines below each mutation profile indicate a wider range of mutation types for which total mutation numbers are different from wild-type. Three stars indicate genotypes with significantly different rates of substitutions, indels or SVs compared to wild-type (FDR < 5%). **B.** Proportion of SVs and indels with breakpoint microhomology across DSBR mutants. **C.** Distribution of microhomology sizes at indel and SV breakpoints across DSBR mutants. **D.** Proportion of SVs with breakpoints in repetitive regions across wild-type and genotypes with elevated SV rates. Dotted lines represent the fraction of variants expected to arise in repetitive regions by chance.
(EPS)

**S5 Fig. Features of structural variants in wild-type, *atm-1*, *brc-1 brd-1*, and *helq-1* mutants.** The number of lines grown for both F20 and/or F40 generations is indicated below each genotype. The presence or absence of repetitive or G4-forming sequences at breakpoints of SVs is indicated by a plus **'+'** 'or minus '-', respectively. A single minus sign '-' indicates that neither of the two breakpoints of the SV occurs in the indicated sequence context. '+/-' indicates repetitive or G4-forming sequences at the left but not the right breakpoint (consistent with the direction on the reference genome chromosome), and '-/+' indicates that such sequences only occur at the right breakpoint. The size and nature of SVs is shown on a log scale on the right. SV are classed into interchromosomal rearrangements (INTCHR), large deletions (DEL), tandem duplications (TD), and inversions (INV). Details on CD Sample IDs and their respective European Nucleotide Archive (ENA) accession numbers are available in the sample description of S1 Table.
(EPS)

**S6 Fig. Features of structural variants in wild-type, *rip-1*, *rfs-1*, *smc-5*, and *smc-6* HR mutants.** The number of lines grown for both F20 and/or F40 generations is indicated below each genotype. The presence or absence of repetitive or G4-forming sequences at breakpoints of SVs is indicated by a plus **'+'** 'or minus '-', respectively. A single minus sign '-' indicates that neither of the two breakpoints of the SV occurs in the indicated sequence context. '+/-' indicates repetitive or G4-forming sequences at the left but not the right breakpoint (consistent with the direction on the reference genome chromosome), and '-/+' indicates that such sequences only occur at the right breakpoint. The size and nature of SVs is shown on a log scale on the right. SV are classed into interchromosomal rearrangements (INTCHR), large deletions (DEL), tandem duplications (TD), and inversions (INV). Details on CD Sample IDs and their respective ENA accession numbers are available in the sample description of S1 Table.
(EPS)

**S7 Fig. Translocations in *rip-1* mutants.** Visual representation of the 3 duplications/translocation type events observed in two *rip-1* F40 lines. CD numbers reflect the European Nucleotide Archive designation of sequenced samples. Translocations (TRLS) are shown using the wild-type genomic region as reference, nucleotides demarking key features, such as the

position of the translocation, the donor sequence and the deletion in the acceptor locus being indicated. In the respective lower panes the level of sequence homology between donor and acceptor loci is indicated. Green arrows indicate tandem duplication-like (TD-like) break-points, blue arrows deletion-like (DEL-like) breakpoints.
(EPS)

**S8 Fig. Features of structural variants in wild-type, *mus-81*, *slx-1*, and *rtel-1* HR mutants.** The number of lines grown for both F20 and/or F40 generations is indicated below each geno-type. The presence or absence of repetitive or G4-forming sequences at breakpoints of SVs is indicated by a plus '**+**' 'or minus '-', respectively. A single minus sign '-' indicates that neither of the two breakpoints of the SV occurs in the indicated sequence context. '+/-' indicates repet-itive or G4-forming sequences at the left but not the right breakpoint (consistent with the direction on the reference genome chromosome), and '-/+' indicates that such sequences only occur at the right breakpoint. The size and nature of SVs is shown on a log scale on the right. SV are classed into interchromosomal rearrangements (INTCHR), large deletions (DEL), tan-dem duplications (TD), and inversions (INV). Details on CD Sample IDs and their respective ENA accession numbers are available in the sample description of S1 Table.
(EPS)

**S9 Fig. Features of structural variants in wild-type, *him-6*, *wrn-1*, and *dog-1* HR mutants.** The number of lines grown for both F20 and/or F40 generations is indicated below each geno-type. The presence or absence of repetitive or G4-forming sequences at breakpoints of SVs is indicated by a plus '**+**' 'or minus '-', respectively. A single minus sign '-' indicates that neither of the two breakpoints of the SV occurs in the indicated sequence context. '+/-' indicates repet-itive or G4-forming sequences at the left but not the right breakpoint (consistent with the direction on the reference genome chromosome), and '-/+' indicates that such sequences only occur at the right breakpoint. The size and nature of SVs is shown on a log scale on the right. SV are classed into interchromosomal rearrangements (INTCHR), large deletions (DEL), tan-dem duplications (TD), and inversions (INV). Details on CD Sample IDs and their respective ENA accession numbers are available in the sample description of S1 Table.
(EPS)

**S10 Fig. Mutational signatures and characteristics of mutagenesis in TLS mutants. A.** Mutational signatures of all tested TLS mutants displayed in numbers of mutations per genera-tion. Bright bars denote individual mutation classes for which numbers differ significantly from wild-type. Bold coloured lines below each mutation profile indicate a wider range of mutation types for which total mutation numbers are different from wild-type. Three stars indicate genotypes with significantly different rates of substitutions, indels or SVs compared to wild-type (FDR < 5%).
(EPS)

**S11 Fig. Mutational signatures and characteristics of mutagenesis in DNA crosslink repair mutants. A.** Mutational signatures of tested DNA crosslink repair mutants displayed in num-bers of mutations per generation. Bright bars denote individual mutation classes for which numbers differ significantly from wild-type. bold coloured bars below each mutation profile indicate a wider range of mutation types for which the total numbers of mutations are different from wild-type. Three stars indicate genotypes with significantly different rates of substitu-tions, indels or SVs compared to wild-type (FDR < 5%). **B.** Distribution of mutations across 11 samples of F20 *dog-1* helicase deficient lines. Over 60% of variants occur in G4-forming regions (large symbols).
(EPS)

**S12 Fig. Possible mechanism for inducing tandem duplications in loci containing inverted repeat sequences.** Step-wise representation of a possible mechanism leading to tandem duplications (TDs) in *helq-1* mutants. Inverted sequences are depicted by red lines with arrows indicating directionality. Replication forks are shown as splayed structures with lagging strand discontinuity indicated by arrows with dotted lines. Scissors indicate a putative endonuclease cleavage site.
(EPS)

**S13 Fig. Telomere proximal structural variants observed in F20 generation *atm-1* mutants.** Structural variants in telomere proximal regions are shown for the indicated *atm-1* F20 lines (S1 Table) with their chromosomal locations. Individual structural variants are coloured based on their classification as duplications (DUP), deletions (DEL), inversion (INV), and interchromosomal rearrangement (BND, breakends) with vertical lines denoting breakpoint locations and horizontal lines spanning the regions between related breakpoints. Black dots represent the coverage in bins of 50 bp or 100 bp (for regions above 20 kb) adjusted for the coverage in the corresponding control sample and indicate copy number.
(EPS)

**S14 Fig. Mutational signatures and characteristics of mutagenesis in checkpoint and apoptosis mutants.** Mutational signatures of the indicated genotypes are displayed in numbers of mutations per generation. Bright bars denote individual mutation classes for which the numbers differ significantly from wild-type. Bold coloured bars below each mutation profile indicate a wider range of mutation types in which the total number of mutations is different from wild-type. Three stars indicate genotypes with significantly different rates of substitutions, indels or SVs compared to wild-type (FDR < 5%).
(EPS)

**S1 Table. Details of *C. elegans* strains used in and sequencing samples from this study.**
(XLSX)

**S2 Table. Analysis of tandem duplications in *helq-1* mutants and their correlation with inverted repeat sequences.**
(XLSX)

**S1 File.**
(DOCX)

## Acknowledgments

We thank the Mitani Lab and the Caenorhabditis Genetics Center for providing *C. elegans* strains. We are grateful to the COMSIG consortium for discussing unpublished data, and to Kei-ichi Takeda and Orlando Schaerer for feedback and proofreading.

## Author Contributions

**Conceptualization:** Nadezda V. Volkova, Moritz Gerstung, Anton Gartner.

**Data curation:** Nadezda V. Volkova, Moritz Gerstung.

**Formal analysis:** Bettina Meier, Nadezda V. Volkova, Moritz Gerstung, Anton Gartner.

**Funding acquisition:** Moritz Gerstung, Anton Gartner.

**Investigation:** Bettina Meier, Nadezda V. Volkova, Ye Hong, Simone Bertolini, Víctor González-Huici, Tsvetana Petrova, Simon Boulton, Peter J. Campbell, Moritz Gerstung, Anton Gartner.

**Project administration:** Moritz Gerstung, Anton Gartner.

**Software:** Moritz Gerstung.

**Supervision:** Moritz Gerstung, Anton Gartner.

**Validation:** Moritz Gerstung.

**Visualization:** Moritz Gerstung.

**Writing – original draft:** Bettina Meier, Nadezda V. Volkova, Moritz Gerstung, Anton Gartner.

**Writing – review & editing:** Moritz Gerstung, Anton Gartner.

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
