## [Decision Letter · Decision Letter 0]

27 Jan 2021

PONE-D-20-39672

Protection of the C. elegans germ cell genome depends on diverse DNA repair pathways during normal proliferation

PLOS ONE

Dear Dr. Gartner,

Thank you for submitting your manuscript to PLOS ONE. After careful consideration, we feel that it has merit but does not fully meet PLOS ONE’s publication criteria as it currently stands. Therefore, we invite you to submit a revised version of the manuscript that addresses the points raised during the review process.

The changes indicated below are required:Statistical analysis: Please indicate how the limits of sequencing coverage influence the identification of homozygous vs heterozygous mutations. Data interpretation: 1.The major challenge is the authors' explanation of the lack of mutagenic effects in NHEJ relative to HR, a result that seems to differ with previous results. This needs to be more clearly described. 2. please discuss the possibility of redundancy within the HR pathways 3. The data consistent with  DNA polymerase slippage may also be explained by sister chromatid recombination.Presentation: Ideally, a better separation of Results and Discussion would benefit the paper.  I understand that a full separation would lead to a lowered comprehension of the extensive data. However, please remove the redundancies present between these two sections. Also please remove any table from the Introduction.Critiques of the four reviewers: Please address all of the issues that are raised by the Reviewers with the exception of those regarding the presentation which is summarized in the previous point. No major rewrite is required.Conflicts between the reviewers: There are no conflicts between the reviewers; just differences in the emphasis of the critique.AE Feedback: My independent reading of the manuscript led to a similar view as the other Reviewers. Of particular importance, I found the explanation for the lack of mutations generated in nhej mutations somewhat lacking. In addition, the results explained on pg. 7 as due to polymerase slippage may  also be explained by sister chromatid exchange.

We look forward to receiving your revised manuscript.

Kind regards,

Arthur J. Lustig, PhD

Academic Editor

PLOS ONE

Journal Requirements:

2.) We noted in your submission details that a portion of your manuscript may have been presented or published elsewhere.

"The majority of the primary sequence data described in this manuscript was also used in our recent publication 'Mutational signatures are jointly shaped by DNA damage and repair. Nat Commun. 2020;11(1):2169. doi: 10.1038/s41467-020-15912-7.

Our Nat Commun. publication is largely focused on studying the effects of treatment with DNA damaging agents in wild-type and DNA repair deficient C. elegans (and only in Fig 1C a high level summary of mutagenesis for 6 mutants is shown). In this submitted manuscript (6 main Figures) we focus on describing mutation rates and patterns of wild-type and 61 repair defective lines when propagated over multiple generations without mutagen exposure. Including primary data in the previously published manuscript was necessary to determine a baseline of DNA damage-induced mutagenesis."

Please clarify whether this publication was peer-reviewed and formally published. If this work was previously peer-reviewed and published, in the cover letter please provide the reason that this work does not constitute dual publication and should be included in the current manuscript.

3.) Please include captions for your Supporting Information files at the end of your manuscript, and update any in-text citations to match accordingly. Please see our Supporting Information guidelines for more information: http://journals.plos.org/plosone/s/supporting-information.

Reviewers' comments:

Reviewer's Responses to Questions

**Comments to the Author**

1. Is the manuscript technically sound, and do the data support the conclusions?

Reviewer #1: Yes

Reviewer #2: Yes

Reviewer #3: Yes

Reviewer #4: Yes

2. Has the statistical analysis been performed appropriately and rigorously? 

Reviewer #1: Yes

Reviewer #2: Yes

Reviewer #3: Yes

Reviewer #4: I Don't Know

3. Have the authors made all data underlying the findings in their manuscript fully available?

Reviewer #1: Yes

Reviewer #2: Yes

Reviewer #3: Yes

Reviewer #4: Yes

4. Is the manuscript presented in an intelligible fashion and written in standard English?

Reviewer #1: Yes

Reviewer #2: Yes

Reviewer #3: Yes

Reviewer #4: Yes

5. Review Comments to the Author

Reviewer #1: The manuscript from Meier and colleagues describes a systematic DNA sequencing analysis of 61 DNA repair mutants maintained in a laboratory setting. The paper follows up on earlier work describing the effects of various mutagens on DNA sequence in the wildtype genome. Here, the authors report the effects of loss of various DNA repair components on the accumulation of different types of DNA damage under mild laboratory conditions where individuals are not subject to mutagenic agents. The study is a very useful catalog of observed sequence changes. It provides information on the frequency and distribution of various mutation types. Interestingly, loss of some DNA repair factors did not impact the mutation rate in the laboratory setting and others correlated with relatively mild increases in mutation rate. The section of the Discussion dealing with the likelihood of mutations arising during meiosis is an important addition. Overall, the methods are thorough and the data are likely to be very useful for researchers studying DNA repair.

Minor comments

Page 3, line 4: substitute “difficult” for “testing” so the sentence reads “Measuring mutation rates in organisms not challenged with mutagens is difficult given the low numbers of mutations.”

Page 5, line 2 from the bottom: substitute a comma for the period after labour-intensive, so the sentence reads “As large-scale investigations of germline mutagenesis are highly time- and labour-intensive, C. elegans offers a suitable system to study mutation accumulation across multiple generations and genetic backgrounds based on its short life-cycle and its ability to self-fertilize.”

Page 7, paragraph 2, line 2: Remove 2nd “across” so the sentence reads “Across 61 C. elegans DNA repair deficient mutants ([10–12], Supplementary Table S1), the median mutation rate was close to that observed in wild-type:…”

Page 11, lines 17-21: The wording of this sentence is awkward so that the meaning is not clear. To clarify, maybe add “including” so the sentence reads: “We deduced that these events involved templated insertions of 200-4000 bp sequences, which showed strong homology to multiple genomic regions, including to one such homeologous region in cis located as far as 275 kb away from the donor sequence on the same chromosome, accompanied by a deletion of several hundred basepairs at the homeologous acceptor site (Supplementary Figure S7).”

Page 19, line 7: It’s confusing to refer to the “latter” in a list of more than two things. Substitute “last” for “latter” so the sentence reads: “Moreover, in ATM deficient yeasts, C.

elegans and Drosophila, the last depending on retrotransposon transposition rather

than telomerase activity for telomere maintenance, chromosome fusions have been

observed cytologically or through sequencing of PCR products across chromosomes

[71–76].”

Page 22, 1st paragraph of Discussion, and elsewhere in the text: “Data” is a plural word, e.g., “Our data provide a comprehensive picture…”

Reviewer #2: The manuscript by Meier et al describes in detailed the natural mutation signature of 60 mutants in various DNA repair pathways. This work is based on the study some of these authors published last year (Volkova et al, ref 10). The current study expanded the analysis from 53 to 61 genetic backgrounds and performed new analysis that was not present in Volkova et al. The detailed analysis provides interesting insights about mutagenesis in an organism under natural, non-mutagenic conditions. Despite the fact that some of the analysis for most of these mutants was done in other studies, there is an important value in doing a large-scale genomic comprehensive study with the same methodology in one publication. Therefore, I believe this analysis will have significant importance to the field. Below are some suggestions for improving the text and figures.

• Abstract. The transition from the first to the second sentence is very abrupt.

• Author Summary. I did not understand this sentence: “Measuring mutation rates in organisms not challenged with mutagens is testing, given low numbers of mutations”.

• Introduction: please remove the reference for data (Sup table 1) and figures form the introduction (it better placed in the results).

• Results: It will be helpful to have more information about how the assay is performed in the beginning of the results section. Although this is based on previously described methodology (ref 10), it will be better to explain this here as well (to the reader who is not familiar with these studies and/or C. elegans). The 2 sentences in the intro (page 6) discussing 1a starting with “In mutation accumulation experiments”, can be moved to the “results”. In addition, 1a referees to 20/40 generations, but some of the mutants were analyzed with less (1 or 5 generations). How mutation accumulation studies done for strains that show embryonic lethality (e.g., smc-5, smc-6) should be described earlier in the text, when the assays are first described. It will be helpful to describe how the 61 DNA repair deficient mutants were selected for the study- what were the criteria?

• Figure 1C- while point mutation are found at similar levels in wild type T>A (with A and T 5’ and 3’ bases) seem to be enriched (to similar levels as 1bp insertions), I don’t think that was reported in other studies- Is this not significant?

• SVs and SNV are defined twice (page 5 and 7)

• Page 9 “showed increased mutation rates without overt changes in mutational signatures…” based on mutation types underlined in figure 3A, it does look like there is mutation signature for base substitutions (with slight variation between mutants in the same pathway) that requires correction and explanation of why this specific pattern

• Page 9. The sentence stating with “comparison between…” is too long and requires more explaining (“COSMIC signature” SBS8…SBS5) for clarity.

• There are several places in the results where it seems would better be moved to the discussion. For example: Page 12 “given the role…” Page 14 the 2 sentences stating with “Clusters of mutations may arise through…” page 17-18 the whole paragraph starting with “At present, we can only speculate how these tandem...”.

• Page 14- “In addition, the NHEJ or MMEJ error-prone DSB repair pathways can also generate clustered mutations when DNA strands with incompatible ends are joined together” yet cku-80 and lig-4 did now show decrease in mutation (or insertions)- please discuss- how come?

• Figure 4C why the difference between smc-5 and smc-6 mutants in deletion size obtained when they are part of the same complex?

• In the text it is stated (page 10): “MMEJ component polq-1, did not produce significant and reliable changes in mutation rates compared to wild-type” and also in figure S4 polq-1 is not significant- while in figure 2 and it is S10 is it significantly different from wild type- is it or is it not significant?

• Page 22 “brc-1/Brca1 and rad-51 paralog mutants show elevated mutagenesis across all types of mutations” bases on 2A it’s not all but most types of mutations.

• It is not clear how HR mutants would lead to deletions, which may be done better in the discussion. With mus-81/slx-1 how D- loop falling apart would lead to loss of genetic material? It makes sense that when HR fails other repair pathways (SSA and MMEJ?) act leading to large deletions but I don’t think it’s explicitly said.

• Discussion and results page 11 and 14: it is not clear why when HR is compromised, base

substitutions are increased and why not all but only particular ones (why specific signature)? Is the signature related to specific TLS polymerases (which ones can be candidates)? Why do TLS polymerases act when HR is compromised or overactive (is this due to aberrant HR or the activation of an alternative repair pathway)?

• Discussion: Page 23- “Thus, the absence of significant mutagenic effects …underpins a high level of redundancy among different DNA repair pathways...” For HR, the mutants that were examined are in genes that only partially required for HR (understandably, since removing HR leads to embryonic lethality). Therefore, HR may be not redundant with other pathways, but the role of HR just cannot be examined. The data may reflect redundancy within HR, not between HR and other pathways.

• Discussion page 25 “It is tempting to speculate that these proteins may not have a

role in HR pathways directly linked with DNA replication” this sentence is not clear to me. Why “tempting to speculate” when a role for most of these proteins in meiotic HR is established?

• Page 26 “Finally, it is reasonable to suggest that many lesions we observe to accumulate in our transgenerational set-up occur in haploid germ cells, especially during meiosis. Indeed, it appears plausible that many SVs might be associated with meiotic recombination.” These 2 sentences don’t seem do lead from one to the other… Germ cells are haploid only after meiotic recombination, for a relatively short period (since recombination occur in meiotic prophase I when the divisions didn’t yet happen). Are these 2 different ideas or should the word “haploid” be removed?

Figures and tables

• Fig 1B- I don’t see a gray line on the diagram (SVs). If this not present in these samples should it be removed from the key to avoid confusion?

• Fig 1C- it’s very hard to see the histograms. I suggest adjusting the y axis to 0.2 max

• Fig 2- 95% CI intervals are truncated for SV plot, maybe extend Y axis further (<0.005?)

• Fig 2- since DSBR is used also to name HR sub pathway, it is confusing to include NHEJ proteins under “DSBR” on the labeling on the top and I suggest labeling cku-80 and lig-4 separately as cNHEJ and use “HR” instead “DSBR” for the others.

• In all the figures that represent mutation spectrum (like 2A) ¬¬- the authors used bars underlining what part of the comparison is significant. However, the sizes of the bars are not consistent and sometimes hard to distinguish from non-significant marking. Sometimes the gray bars are invisible.

• S2- what are the pink and orange boxes in partp-1 and parp-2 respectively?

• S7- italics for gene names

• S10 and S14- X axis is not clear, letter on top of each other.

• Table 1- Key for “DNA damage response pathway”, why “Helicase, TR” in green?

Reviewer #3: 6th Jan 2021

The authors systematically catalogued the mutational characteristics of DNA repair

deficiencies across the C. elegans DNA repair and damage response

pathways in animals that were cultivated to 40 generations.

They provide new information on the contributions of different DNA repair pathways towards mutagenesis that was not previously known. For example, they showed that

the DOG-1/FANCJ helicase has a unique function to unwind G-quadruplex forming sequences to prevent mutations. Likewise, they showed that the ATM-1 checkpoint kinase perform a specific role in avoiding mutations in the sub-telomeric repeats from

DSBs. The authors should be commended for a valuable and elaborate study.

Comment:

The article is excessively long (result section in particular) and the authors could shortened it to enhance readership by taking the gene deletion mutant that showed no effect and place these in a Table in the text. In addition, they could give a brief description of the gene function. Reading one section to the next gives the same pattern of information, except for focusing on a different pathway, which makes reading the results section somewhat redundant and less appealing.

Reviewer #4: The study by Meier et al. reports the mutational profile of 61 C. elegans DNA repair mutants propagated 40 generations in the absence of exogenous stress. The article is very dense but in general clearly presented.

The authors find that across DNA repair deficient mutants, the median mutation rate is close to wild-type. However, 42 out of the 61 DNA repair deficient strains analyzed displayed mutation rates significantly different from wild-type in at least one mutation class (base substitutions, indels or stuctural variants). Characterization of the mutational signatures over generations in DNA repair pathways deficient in DR, BER, and NER did not show large changes in the overall mutation spectra, but several small differences in particular mutation types. Inactivation of the core components of NHEJ did not produce significant changes in mutation rates compared to wild-type, while mutants in homologous recombination (HR) showed either increased mutations across all mutation classes, or primarily accumulated structural variants. The discussion speculated of possible mechanisms explaining the specific mutational signature observed, and relate these to the biology of C. elegans.

Collectively the data presented constitutes a useful resource for future studies on how conserved DNA repair pathways contribute to preserving germline genome integrity and is appropriate for publication in PloS One.

Specific comments:

1. Please specify from what stage DNA was isolated. Is is young adults?

2. The 10X coverage in these experients is on the low end on the scale, and while adequate for calling homozygous variants, may not be not be enough to accurately call structural variants or heterozygous mutations. The authors could mention this caveat somehere. Nonetheless this data provides may surve as a basis for future, more in depth studies.

3. Fig 1C: Single nucleotide variants are shown in the 'context of their 5’ and 3’ base'.

can authors please reformulate? 5' and 3' with respect to what?

General comments:

The study relies heavily on analysis of genome sequencing data. Methods are described in detail but this reviewer does not have the competences to judge whether the statistical methods and other analyses were adequately performed. All data is available in supplementary tables or from appropriate databases using provided accession numbers.

6. PLOS authors have the option to publish the peer review history of their article (what does this mean?). If published, this will include your full peer review and any attached files.

Reviewer #1: No

Reviewer #2: No

Reviewer #3: **Yes: **Dindial Ramotar

Reviewer #4: No

---

## [Author Response · Author response to Decision Letter 0]

3 Mar 2021

Response to reviewers:

We think that is fair to say that all 4 reviewers strongly supported our manuscript and we are grateful for the comments that helped improve our manuscript (see our point by point response). 

Following your and the reviewers guidance we proofread the manuscript again and think that it is now easier to read. We incorporated your and the reviewers comments as described in our point-by-point response appended below. While we still describe our results, pathway by pathway, we moved the more detailed discussion to the discussion section following your and the reviewers advice. We also formatted the manuscript according to the PLOS ONE format guidelines (9efad3c44db3a7f319fbf03644b95f8f986e8e6).

Two important points:

A) Reviewer 2 pointed out that our data indicates that ‘..fcd-2/FANCD2 and fnci-1/FANCI ICL repair defective lines, and polq-1 polymerase theta/POLQ (microhomology mediated end-joining) show reduced rates of indels and single nucleotide variants (Polq). 

However we stand by our statement interpreting these data which are described in Figure 2 as follows 

‘Several DNA repair mutants, namely fcd-2/FANCD2 and fnci-1/FANCI DNA ICL-repair defective lines, and the microhomology mediated end-joining (MMEJ) defective polq-1/POLQ mutant exhibited reduced indel rates (Fig 2, green dots) Additionally, polq-1 mutants harboured reduced SNVs (Fig 2, red dot). POLQ-1 dependent MMEJ is an error prone pathway, in which resected 3’ single-stranded overhangs pair at their complementary terminal nucleotide(s) to prime DNA synthesis, often leading to small deletions [15–18]. However, given that indel and SV mutation rates in our dataset are already low in wild-type and more wild-type than mutant samples were included in the analysis (wild-type n = 30, mutant n = 4-8) the sample variance in genotypes with mutation rates close or lower to wild-type may be underestimated. We therefore caution that the observed reductions in mutagenesis levels are likely to be false discoveries. ‘

We hope that the changes we have made to this paragraph clarifies our reasoning.

B) Reviewer 4 asked if 10 time sequence coverage allows for accurately calling structural variants and heterozygous mutations. 

We apologise for mistakenly stating that the sequencing coverage was only 10 fold. Indeed, sequencing coverage across our analysis was around 50 fold. We have corrected our mistake and improved the methods sections. Additionally we now refer to the detailed methods collection which we published as part of our previous study Ref. 10, Volkova et al., Nat Commun. 2020;11: 2169, and which also provides a detailed description of the bioinformatics pipeline. 

Finally responding to the note from PLOS ONE that ‘We noted in your submission details that a portion of your manuscript may have been presented or published elsewhere and the request to ‘Please clarify whether this publication was peer-reviewed and formally published. If this work was previously peer-reviewed and published, in the cover letter please provide the reason that this work does not constitute dual publication and should be included in the current manuscript’

As we indicated in our initial submission we largely use the same primary data as used in a previously published and reviewed paper Nat Commun. 2020;11(1):2169. doi: 10.1038/s41467-020-15912-7. It is common in the area of bioinformatics that primary source data are used in new publications as we did. Our previous Nat Commun. publication is largely focused on studying the effects of treatment with DNA damaging agents in wild-type and DNA repair deficient C. elegans. In the previous paper only Fig 1C provides a high level summary of background mutagenesis (6 mutants are shown). In our PLOS ONE manuscript we describe in detail the mutations arising in wild-type and 61 repair mutants when propagated over multiple generations without mutagen exposure. This description comprises 6 Figures and 14 Supplementary Figures describing in detail mutation rates and patterns of wild-type and 61 repair defective lines when propagated over multiple generations. In addition, we describe sequencing data from 5 double mutants. All 4 reviewers support our PLOS ONE publication supporting our case that the in-depth analysis of effects of DNA repair deficiency analysis over generations presented here exceeds our previous work published in Nat Commun.

Below please find a point-by point-response to the reviewers and the associate editor. We upload two text files, one where changes we made are indicated, the other one, where these changes are accepted.

Best regards,

Anton Gartner and Moritz Gerstung

5. Review Comments to the Author

Reviewer #1: The manuscript from Meier and colleagues describes a systematic DNA sequencing analysis of 61 DNA repair mutants maintained in a laboratory setting. The paper follows up on earlier work describing the effects of various mutagens on DNA sequence in the wildtype genome. Here, the authors report the effects of loss of various DNA repair components on the accumulation of different types of DNA damage under mild laboratory conditions where individuals are not subject to mutagenic agents. The study is a very useful catalog of observed sequence changes. It provides information on the frequency and distribution of various mutation types. Interestingly, loss of some DNA repair factors did not impact the mutation rate in the laboratory setting and others correlated with relatively mild increases in mutation rate. The section of the Discussion dealing with the likelihood of mutations arising during meiosis is an important addition. Overall, the methods are thorough and the data are likely to be very useful for researchers studying DNA repair.

Minor comments

Page 3, line 4: substitute “difficult” for “testing” so the sentence reads “Measuring mutation rates in organisms not challenged with mutagens is difficult given the low numbers of mutations.”

thanks, done

Page 5, line 2 from the bottom: substitute a comma for the period after labour-intensive, so the sentence reads “As large-scale investigations of germline mutagenesis are highly time- and labour-intensive, C. elegans offers a suitable system to study mutation accumulation across multiple generations and genetic backgrounds based on its short life-cycle and its ability to self-fertilize.

thanks, done

Page 7, paragraph 2, line 2: Remove 2nd “across” so the sentence reads “Across 61 C. elegans DNA repair deficient mutants ([10–12], Supplementary Table S1), the median mutation rate was close to that observed in wild-type:…”thanks, done

Page 11, lines 17-21: The wording of this sentence is awkward so that the meaning is not clear. To clarify, maybe add “including” so the sentence reads: “We deduced that these events involved templated insertions of 200-4000 bp sequences, which showed strong homology to multiple genomic regions, including to one such homeologous region in cis located as far as 275 kb away from the donor sequence on the same chromosome, accompanied by a deletion of several hundred basepairs at the homeologous acceptor site (Supplementary Figure S7).”

thanks, done

Page 19, line 7: It’s confusing to refer to the “latter” in a list of more than two things. Substitute “last” for “latter” so the sentence reads: “Moreover, in ATM deficient yeasts, C.

elegans and Drosophila, the last depending on retrotransposon transposition rather

than telomerase activity for telomere maintenance, chromosome fusions have been

observed cytologically or through sequencing of PCR products across chromosomes

[71–76].

thanks, done

Page 22, 1st paragraph of Discussion, and elsewhere in the text: “Data” is a plural word, e.g., “Our data provide a comprehensive picture…”

thanks, done throughout the manuscript.

Reviewer #2: The manuscript by Meier et al describes in detailed the natural mutation signature of 60 mutants in various DNA repair pathways. This work is based on the study some of these authors published last year (Volkova et al, ref 10). The current study expanded the analysis from 53 to 61 genetic backgrounds and performed new analysis that was not present in Volkova et al. The detailed analysis provides interesting insights about mutagenesis in an organism under natural, non-mutagenic conditions. Despite the fact that some of the analysis for most of these mutants was done in other studies, there is an important value in doing a large-scale genomic comprehensive study with the same methodology in one publication. Therefore, I believe this analysis will have significant importance to the field. Below are some suggestions for improving the text and figures.

• Abstract. The transition from the first to the second sentence is very abrupt.

Thanks, we took on this point and revised the beginning of the abstract which now reads as follows.

‘Maintaining genome integrity is particularly important in germ cells to ensure faithful transmission of genetic information across generations. Here we systematically describe germ cell mutagenesis in wild-type and 61 DNA repair mutants cultivated over multiple generations. ~44% of the DNA repair mutants analysed showed a >2-fold increased mutagenesis with a broad spectrum of mutational outcomes.’

• Author Summary. I did not understand this sentence: “Measuring mutation rates in organisms not challenged with mutagens is testing, given low numbers of mutations”.

Thanks, we changed this sentence to ‘Measuring mutation rates in organisms not challenged with mutagens is difficult, given low numbers of mutations’

• Introduction: please remove the reference for data (Sup table 1) and figures form the introduction (it better placed in the results).

done

• Results: It will be helpful to have more information about how the assay is performed in the beginning of the results section. Although this is based on previously described methodology (ref 10), it will be better to explain this here as well (to the reader who is not familiar with these studies and/or C. elegans). The 2 sentences in the intro (page 6) discussing 1a starting with “In mutation accumulation experiments”, can be moved to the “results”. In addition, 1a referees to 20/40 generations, but some of the mutants were analyzed with less (1 or 5 generations). How mutation accumulation studies done for strains that show embryonic lethality (e.g., smc-5, smc-6) should be described earlier in the text, when the assays are first described. It will be helpful to describe how the 61 DNA repair deficient mutants were selected for the study- what were the criteria?

Thanks: We have restructured the end of the introduction and the beginning of the result section to better describe our methodology upfront. This now better guides the non-expert reader and together with Figure 1 the methodology is made clear up front. Also, we improved the relevant methods section, and in addition better refer to our detailed methods section provided in Ref 10 (Volkova, Meier et al., 2020). We now also, both in the relevant results as well as in the methods section, refer to Suppl Table 1 where the number of lines sequenced for each strain is listed, as is the number of generations. In the methods section we now also state that rtel-1, smc-5 andsmc-6 mutants were only grown for 5 generations given that many of the lines grown in parallel had already become sterile at generation 5. 

• Figure 1C- while point mutation are found at similar levels in wild type T>A (with A and T 5’ and 3’ bases) seem to be enriched (to similar levels as 1bp insertions), I don’t think that was reported in other studies- Is this not significant?

Thanks, we have added the information that T>A changes with A and T 5’ and 3’ bases represent the most abundant SNVs. The relevant section now reads ‘Mutations were equally distributed across the wild-type genome with no evidence of clustering (Figure 1B). The most frequent mutations were a) single base insertions, with T>A changes in the context of a 5’A and a 3’ T and b) deletions in homopolymeric sequences (Figure 1C).’

• SVs and SNV are defined twice (page 5 and 7)

Thanks, we have removed these duplications.

• Page 9 “showed increased mutation rates without overt changes in mutational signatures…” based on mutation types underlined in figure 3A, it does look like there is mutation signature for base substitutions (with slight variation between mutants in the same pathway) that requires correction and explanation of why this specific pattern

We want to be cautious in this case, focusing on where data is firm. Changes in signatures across NER mutants do not appear as uniform enough to make strong statements. This is why we say ‘without overt changes’. In other words, while our data clearly supports increased mutagenesis in NER defective lines, working our signatures changes associated with different NER repair mutants would need analyzing a larger number of analysed worm lines.

• Page 9. The sentence stating with “comparison between…” is too long and requires more explaining (“COSMIC signature” SBS8…SBS5) for clarity.

Thanks, we reworded this section which is now easier to read.

• There are several places in the results where it seems would better be moved to the discussion. For example: Page 12 “given the role…” Page 14 the 2 sentences stating with “Clusters of mutations may arise through…” page 17-18 the whole paragraph starting with “At present, we can only speculate how these tandem...”.

OK, we relooked at both sections, and would like to make the case for leaving the structure of the narrative as is in two cases Page 12 and 14. The tricky thing about our paper is us describing many genotypes, each requiring an introduction. This is why we ordered the results section pathway per pathway, in each case providing a short introduction and at the end a short discussion (eg introduction-experiment-conclusion). We think that this format is more readable. Concerning our discussion page 17-18 relating to how tandem duplications might arise in helq mutants we agree and shifted this entire section to the discussion. 

• Page 14- “In addition, the NHEJ or MMEJ error-prone DSB repair pathways can also generate clustered mutations when DNA strands with incompatible ends are joined together” yet cku-80 and lig-4 did now show decrease in mutation (or insertions)- please discuss- how come?

Thanks. We added the following speculative sentence, “The absence of clustered mutations in NHEJ or MMEJ defective lines could be explained by the action of redundant, error-free HR pathways.”

• Figure 4C why the difference between smc-5 and smc-6 mutants in deletion size obtained when they are part of the same complex?

The plots in Figure 4C lower panel clearly show that average deletion size observed in smc-5 and smc-6 is similar. There is a difference in the size of tandem duplication (TD), however the number of TDs observed is small.

• In the text it is stated (page 10): “MMEJ component polq-1, did not produce significant and reliable changes in mutation rates compared to wild-type” and also in figure S4 polq-1 is not significant- while in figure 2 and it is S10 is it significantly different from wild type- is it or is it not significant?

Thanks, we now state that we observed a reduced number of SNV and indels in polq-1 mutants. We, however, did not observe a signature change (S10). However as explained to the editor our reasoning (incorporated into the manuscript) is as follows:

‘Several DNA repair mutants, namely fcd-2/FANCD2 and fnci-1/FANCI DNA ICL-repair defective lines, and the microhomology mediated end-joining (MMEJ) defective polq-1/POLQ mutant exhibited reduced indel rates (Figure 2, green dots) Additionally, polq-1 mutants harboured reduced SNVs (Figure 2, red dot). POLQ-1 dependent MMEJ is an error prone pathway, where resected 3’ single-stranded overhangs pair at their complementary terminal nucleotide(s) to prime DNA synthesis, often leading to small deletions. [15–18]. However, given that indel and SV mutation rates in our dataset are already low in wild-type and more wild-type samples were included in the analysis (wild-type n = 30, mutant n = 4-8) the sample variance in genotype showing mutation rates close or lower to wild-type, may be underestimated. We therefore caution that the observed reductions in mutagenesis levels are likely to be false discoveries. ‘

• Page 22 “brc-1/Brca1 and rad-51 paralog mutants show elevated mutagenesis across all types of mutations” bases on 2A it’s not all but most types of mutations.

Thanks, we changed to ‘most’

• It is not clear how HR mutants would lead to deletions, which may be done better in the discussion. With mus-81/slx-1 how D- loop falling apart would lead to loss of genetic material? It makes sense that when HR fails other repair pathways (SSA and MMEJ?) act leading to large deletions but I don’t think it’s explicitly said.

Thanks, we added the following sentence at the end of the HR section in the discussion.

‘An increased number of deletions in HR mutants might arise from the error prone activity of redundant SSA and MMEJ repair pathways 

• Discussion and results page 11 and 14: it is not clear why when HR is compromised, base

substitutions are increased and why not all but only particular ones (why specific signature)? Is the signature related to specific TLS polymerases (which ones can be candidates)? Why do TLS polymerases act when HR is compromised or overactive (is this due to aberrant HR or the activation of an alternative repair pathway)?

Of course, we would like to know the exact answer to this question…

We speculate in the discussion as follows. ….We suspect that increased point mutations might be a scar indicative of error prone translesion synthesis, necessary when damaged bases are neither repaired by BER and NER nor by replication fork reversal which is linked to recombinational repair [95]. Point mutations and small deletions also occur when HR is replaced by more error-prone NHEJ or MMEJ pathways, the latter being associated with the occurrence of small deletions in human BRCA1 mutants [27]. Conversely, deficiencies of other HR proteins, like SLX-1 and MUS-81, and helicases including HIM-6, RTEL-1 and HELQ-1, are associated with a specific increase of SVs. It is tempting to speculate that these proteins may not have a role in HR pathways directly linked with DNA replication (see below). An increased number of deletions in HR mutants might arise from the error prone activity of redundant single strand annealing and MMEJ repair pathways. 

• Discussion: Page 23- “Thus, the absence of significant mutagenic effects …underpins a high level of redundancy among different DNA repair pathways...” For HR, the mutants that were examined are in genes that, only partially required for HR (understandably, since removing HR leads to embryonic lethality). Therefore, HR may be not redundant with other pathways, but the role of HR just cannot be examined. The data may reflect redundancy within HR, not between HR and other pathways.

 Thanks, we reflect this concern and now added ‘within and’ such the sentence reads ….high level of redundancy within and among different DNA repair pathways... ‘

• Discussion page 25 “It is tempting to speculate that these proteins may not have a

role in HR pathways directly linked with DNA replication” this sentence is not clear to me. Why “tempting to speculate” when a role for most of these proteins in meiotic HR is established?

Thanks, we changed to ..’we speculate…’. RT

• Page 26 “Finally, it is reasonable to suggest that many lesions we observe to accumulate in our transgenerational set-up occur in haploid germ cells, especially during meiosis. Indeed, it appears plausible that many SVs might be associated with meiotic recombination.” These 2 sentences don’t seem do lead from one to the other… Germ cells are haploid only after meiotic recombination, for a relatively short period (since recombination occur in meiotic prophase I when the divisions didn’t yet happen). Are these 2 different ideas or should the word “haploid” be removed?

Thanks, corrected, we removed the word ‘haploid’

Figures and tables

• Fig 1B- I don’t see a gray line on the diagram (SVs). If this not present in these samples should it be removed from the key to avoid confusion?

done

• Fig 1C- it’s very hard to see the histograms. I suggest adjusting the y axis to 0.2 max

thanks, done.

• Fig 2- 95% CI intervals are truncated for SV plot, maybe extend Y axis further (<0.005?) 

thanks: CI are truncated because they extend to 0, which is impossible to show on a log scale blot. We added a sentence to mention this in the Figure legend reding. ‘All CIs which extend below the lower edge of the plot have zero as their lower border’

• Fig 2- since DSBR is used also to name HR sub pathway, it is confusing to include NHEJ proteins under “DSBR” on the labeling on the top and I suggest labeling cku-80 and lig-4 separately as cNHEJ and use “HR” instead “DSBR” for the others.

Here we would like to keep organisation as is. NHEJ and all HR subpathways fall into DSBR (double strand break repair). cku-80 and lig-4 are next to each other, at the left to the DSBR section.

• In all the figures that represent mutation spectrum (like 2A) ¬¬- the authors used bars underlining what part of the comparison is significant. However, the sizes of the bars are not consistent and sometimes hard to distinguish from non-significant marking. Sometimes the gray bars are invisible.

Thanks, We have double checked all main and Suppl Figures. The bars are clearly visible when PDF files are opened.

• S2- what are the pink and orange boxes in partp-1 and parp-2 respectively?

thanks, we have fixed this ‘The pink and orange shaded regions in parp-1 and parp-2 samples, respectively, indicate the location and extent of observed structural variants (pink - TD (tandem duplication); orange - Fold (foldback duplication)).’

• S7- italics for gene names

thanks, this is fixed.

• S10 and S14- X axis is not clear, letter on top of each other.

thanks, this is fixed.

• Table 1- Key for “DNA damage response pathway”, why “Helicase, TR” in green?

thanks, this is fixed.

Reviewer #3: 6th Jan 2021

The authors systematically catalogued the mutational characteristics of DNA repair

deficiencies across the C. elegans DNA repair and damage response

pathways in animals that were cultivated to 40 generations.

They provide new information on the contributions of different DNA repair pathways towards mutagenesis that was not previously known. For example, they showed that

the DOG-1/FANCJ helicase has a unique function to unwind G-quadruplex forming sequences to prevent mutations. Likewise, they showed that the ATM-1 checkpoint kinase perform a specific role in avoiding mutations in the sub-telomeric repeats from

DSBs. The authors should be commended for a valuable and elaborate study.

Comment:

The article is excessively long (result section in particular) and the authors could shortened it to enhance readership by taking the gene deletion mutant that showed no effect and place these in a Table in the text. In addition, they could give a brief description of the gene function. Reading one section to the next gives the same pattern of information, except for focusing on a different pathway, which makes reading the results section somewhat redundant and less appealing.

Thanks, as detailed in the response to the above reviewers we now provide a better introduction of the experimental system to non-expert readers, we improved the clarity of our wording in many instances, and also shifted the discussion of how TD may arise in Hel-Q mutants to the discussion section. Overall, we think that our paper now reads much better. 

Reviewer #4: The study by Meier et al. reports the mutational profile of 61 C. elegans DNA repair mutants propagated 40 generations in the absence of exogenous stress. The article is very dense but in general clearly presented.

The authors find that across DNA repair deficient mutants, the median mutation rate is close to wild-type. However, 42 out of the 61 DNA repair deficient strains analyzed displayed mutation rates significantly different from wild-type in at least one mutation class (base substitutions, indels or stuctural variants). Characterization of the mutational signatures over generations in DNA repair pathways deficient in DR, BER, and NER did not show large changes in the overall mutation spectra, but several small differences in particular mutation types. Inactivation of the core components of NHEJ did not produce significant changes in mutation rates compared to wild-type, while mutants in homologous recombination (HR) showed either increased mutations across all mutation classes, or primarily accumulated structural variants. The discussion speculated of possible mechanisms explaining the specific mutational signature observed, and relate these to the biology of C. elegans.

Collectively the data presented constitutes a useful resource for future studies on how conserved DNA repair pathways contribute to preserving germline genome integrity and is appropriate for publication in PloS One.

Specific comments:

1. Please specify from what stage DNA was isolated. Is is young adults?

Genomic DNA was isolated from a mix stage, clonal population (first and second generation) derived from a single ‘final generation” worm. This is now more clearly explained in Materials and Methods. 

2. The 10X coverage in these experiments is on the low end of the scale, and while adequate for calling homozygous variants, may not be not be enough to accurately call structural variants or heterozygous mutations. The authors could mention this caveat somewhere. Nonetheless this data provides may surve as a basis for future, more in depth studies.

Thanks for spotting this. Indeed, our mean coverage is 50X. Data acquisition, mutation calling, filtering and classification is described in great detail in our previous paper (ref 10). We rectified our error and better refer to ref 10. We can thus call structural variants or heterozygous mutations with great confidence.

3. Fig 1C: Single nucleotide variants are shown in the 'context of their 5’ and 3’ base'.

can authors please reformulate? 5' and 3' with respect to what?

These are the nucleotides 5’ and 3’ of the SNV.

General comments:

The study relies heavily on analysis of genome sequencing data. Methods are described in detail but this reviewer does not have the competences to judge whether the statistical methods and other analyses were adequately performed. All data is available in supplementary tables or from appropriate databases using provided accession numbers.

AE Feedback: My independent reading of the manuscript led to a similar view as the other Reviewers. Of particular importance, I found the explanation for the lack of mutations generated in nhej mutations somewhat lacking. In addition, the results explained on pg. 7 as due to polymerase slippage may also be explained by sister chromatid exchange.

We have included a sentence explaining not observing increased mutagenesis end joining mutants might be because of end joining in C. elegans predominantly acting in somatic cells and not in germ cells, a notion supported by previous reports we cite. Of course, in the absence of end-joining HR may take over, a possibility we now also discuss. For discussing the genesis of indels in homopolymeric repeat sequences we refer to our previous in-depth analysis of MMR defective stains which favours a slippage model [11].

---

## [Decision Letter · Decision Letter 1]

25 Mar 2021

PONE-D-20-39672R1

Protection of the C. elegans germ cell genome depends on diverse DNA repair pathways during normal proliferation

PLOS ONE

Dear Dr. Gartner,

Thank you for submitting your manuscript to PLOS ONE. After careful consideration, we feel that it has merit but does not fully meet PLOS ONE’s publication criteria as it currently stands. Therefore, we invite you to submit a revised version of the manuscript that addresses the points raised during 

Only one trivial change. Please delete the worries indicated be Reviewet 4 and resubmittedments here and delete this placeholder text when finished. Be sure 

We look forward to receiving your revised manuscript.

Kind regards,

Arthur J. Lustig, PhD

Academic Editor

PLOS ONE

Journal Requirements:

Reviewers' comments:

Reviewer's Responses to Questions

**Comments to the Author**

1. If the authors have adequately addressed your comments raised in a previous round of review and you feel that this manuscript is now acceptable for publication, you may indicate that here to bypass the “Comments to the Author” section, enter your conflict of interest statement in the “Confidential to Editor” section, and submit your "Accept" recommendation.

Reviewer #2: All comments have been addressed

Reviewer #3: All comments have been addressed

Reviewer #4: All comments have been addressed

2. Is the manuscript technically sound, and do the data support the conclusions?

Reviewer #2: Yes

Reviewer #3: Yes

Reviewer #4: Yes

3. Has the statistical analysis been performed appropriately and rigorously? 

Reviewer #2: Yes

Reviewer #3: Yes

Reviewer #4: Yes

4. Have the authors made all data underlying the findings in their manuscript fully available?

Reviewer #2: Yes

Reviewer #3: Yes

Reviewer #4: Yes

5. Is the manuscript presented in an intelligible fashion and written in standard English?

Reviewer #2: Yes

Reviewer #3: Yes

Reviewer #4: Yes

6. Review Comments to the Author

Reviewer #2: The authors appropriately addressed all my concerns and questions and I have no further reservations or concerns.

Reviewer #3: The authors have now carefully addressed all concerns and I am satisfy with the revisions. The authors have improved the manuscript.

Reviewer #4: The manuscript has been improved and this reviewers comments were taken into account

the phrase "Our study provides an analysis of 528 whole genome sequencing" appears at bottom of each page and should be deleted

7. PLOS authors have the option to publish the peer review history of their article (what does this mean?). If published, this will include your full peer review and any attached files.

Reviewer #2: No

Reviewer #3: **Yes: **Dindial Ramotar

Reviewer #4: No

---

## [Author Response · Author response to Decision Letter 1]

31 Mar 2021

I now uploaded the final manuscript. I removed the footer which reviewer 4 asked us to remove. (NOTE, this change is not indicated in the file that asked me to highlight all changes. Reviewers and the Editors did not ask for any other change

---

## [Editor Report · Decision Letter 2]

5 Apr 2021

Protection of the C. elegans germ cell genome depends on diverse DNA repair pathways during normal proliferation

PONE-D-20-39672R2

Dear Dr. Gartner,

We’re pleased to inform you that your manuscript has been judged scientifically suitable for publication and will be formally accepted for publication once it meets all outstanding technical requirements.

Kind regards,

Arthur J. Lustig, PhD

Academic Editor

PLOS ONE
---

## [Editor Report · Acceptance letter]

14 Apr 2021

PONE-D-20-39672R2 

Protection of the *C. elegans* germ cell genome depends on diverse DNA repair pathways during normal proliferation 

Dear Dr. Gartner:

I'm pleased to inform you that your manuscript has been deemed suitable for publication in PLOS ONE. Congratulations! Your manuscript is now with our production department. 

Kind regards, 

on behalf of

Dr. Arthur J. Lustig 

Academic Editor

PLOS ONE